# Safety-Aligned Weights Are Not Enough: Refusal-Teacher-Guided Finetuning Enhances Safety and Downstream Performance under Harmful Finetuning Attacks

## Abstract

Recently, major AI providers such as Google and OpenAI have introduced Finetuning-as-a-Service (FaaS), which allows users to customize Large Language Models (LLMs) using their own data. However, this service is vulnerable to safety degradation when user data includes harmful prompts, a threat known as harmful finetuning attacks. Prior works attempt to mitigate this issue by first constructing safety-aligned model and then finetuning the model on user data. However, we observe that the safety-aligned weights provide weak initialization for downstream task learning, leading to suboptimal safety-alignment and downstream task performance. To address this, we propose a **Refusal-Teacher (Ref-Teacher)-guided finetuning framework**. Instead of finetuning a safety-aligned model on user data, our approach directly finetunes the base model under the guidance of a safety-aligned Ref-Teacher, which filters harmful prompts from user data and distills safety-alignment knowledge into the base model. Extensive experiments demonstrate that our Ref-Teacher-guided finetuning strategy effectively minimizes harmful outputs and enhances finetuning accuracy for user-specific tasks, offering a practical solution for secure and reliable deployment of LLMs in FaaS.

## 1 Introduction

Recent advancements in Large Language Models (LLMs) (Touvron et al. (2023); Jiang et al. (2023); Team et al. (2024); Team (2024); Hurst et al. (2024); Guo et al. (2025); Research et al. (2025)) have achieved remarkable performance across a wide range of natural language processing tasks. LLMs are typically pretrained on massive and diverse corpora, resulting in strong generalization ability and broad applicability across domains. To further facilitate LLMs for individual and domain-specific purposes, major AI service providers such as Google and OpenAI offer not only access to pretrained LLMs but also Finetuning-as-a-Service (FaaS). This service enables users to upload custom datasets and adapt LLMs to more specific tasks and domains depending on their unique requirements.

However, FaaS must prevent the malicious use of LLMs through safety-alignment, even when users attempt to jailbreak the models via customization. These types of attacks, which inject harmful prompts into user data for finetuning, are called *harmful finetuning attacks*. Several studies (Qi et al. (2023); Lermen et al. (2023); Rosati et al. (2024); Huang et al. (2024b;c;d); Li et al. (2025); Huang et al. (2025)) have shown that finetuning on user data containing harmful content compromises the safety-alignment, despite the LLMs being safety-aligned before finetuning. This vulnerability highlights the need to preserve safety while achieving high performance on user tasks in FaaS.

To mitigate these risks, prior works typically adopt a two-stage pipeline. In the first stage, referred to as the *alignment stage*, pretrained LLMs are trained on safety-alignment data to avoid generating harmful responses. In the second stage, referred to as the *finetuning stage*, the safety-aligned models are finetuned on user data for user-specific downstream tasks. Within this pipeline, some methods find robust model weights against harmful finetuning attacks during the alignment stage (Huang et al. (2024c;d); Liu et al. (2024); Rosati et al. (2024)), while others preserve safety-aligned weights during the finetuning stage (Mukhoti et al. (2023); Huang et al. (2024b); Li et al. (2024a; 2025)).

However, we observe that the two-stage pipeline adopted in prior works is suboptimal. Safety-aligned models provide weak weight initialization for learning downstream tasks, resulting in limited task performance and compromised safety. A more effective alternative is to directly finetune the base model on both user data and safety-alignment data during finetuning stage, thereby enhancing downstream task performance while preserving safety. Nevertheless, this base model finetuning strategy suffers from gradient conflicts between the two objectives, safety and user task, which destabilize training and are further exacerbated when user data contains harmful prompts.

Building on these observations, we propose a novel **Refusal-Teacher (Ref-Teacher)-guided fine-tuning framework** (Fig. 1), which directly finetunes the base model on both user data and safety-alignment data under the guidance of a Ref-Teacher. In our framework, the Ref-Teacher serves two complementary roles. First, it performs **Alignment Distillation** by generating soft refusal labels that provide richer supervision and yield smoother loss surfaces, thereby mitigating gradient conflicts. Second, it performs **Data Filtering** by removing harmful prompts from user data based on its refusal feature, ensuring robust conflict mitigation against harmful finetuning attacks. Through these two roles, our framework effectively alleviates gradient conflicts, which in turn enables improved safety and downstream task performance even under harmful finetuning attacks.

Our extensive experiments demonstrate the effectiveness of the Ref-Teacher-guided finetuning framework in enhancing both user-specific task performance and safety-alignment. Across a wide range of evaluations, our method consistently achieves the highest finetuning accuracy and the lowest harmful scores compared to all baselines. Consequently, our framework overcomes the limitations of prior two-stage pipelines and offers a practical solution for secure and reliable FaaS.

**Our Contributions.**

- We demonstrate that safety-aligned LLMs provide weak initialization for downstream learning, resulting in suboptimal task performance and compromised safety, whereas directly finetuning the base model on safety-alignment data and user data improves both safety and task performance.

- However, this base model finetuning strategy suffers from gradient conflicts between safety and user task objectives, which are further exacerbated when user data includes harmful prompts. To overcome this, we propose the Refusal-Teacher(Ref-Teacher)-guided finetuning framework, which mitigates such conflicts through (i) alignment distillation and (ii) data filtering.

- Extensive experiments demonstrate that our framework achieves strong performance on user-specific downstream tasks while consistently preserving safety across diverse settings.

## 2 RELATED WORKS

**Safety in Large Language Models.** Large Language Models (LLMs) can respond to diverse queries but are vulnerable to harmful prompts (Ji et al. (2023); Zou et al. (2023)), which can elicit unsafe outputs such as weapon-making instructions. To mitigate these risks, safety-aligned LLMs (Team (2024); Llama Team (2024); Team et al. (2024)) have been developed, trained via Supervised Fine-Tuning (Bianchi et al. (2023)) or Reinforcement Learning with Human Feedback (Ouyang et al. (2022); Rafailov et al. (2023)) on datasets that pair harmful prompts with refusal responses, enabling them to reject unsafe requests. Nevertheless, they remain vulnerable to advanced jailbreaking techniques (Chao et al. (2023); Liu et al. (2023); Zou et al. (2023); Li et al. (2024b)). Training-free defenses leverage LLMs' ability to assess harmfulness (Wang et al. (2024); Zhang et al. (2024)), or exploit internal differences when processing harmful versus harmless inputs (Xie et al. (2024); Hu et al. (2024); Hung et al. (2024)). In contrast, training-based methods enhance robustness by finetuning LLMs through adversarial training. Some approaches adjust the balance of harmful and harmless prompts (Bianchi et al. (2023)), while others generate adversarial samples via latent-space perturbations (Sheshadri et al. (2024a;b); Xhonneux et al. (2024); Zou et al. (2024); Yu et al. (2024)).

Other methods train separate safe and unsafe models and apply safe decoding strategies Banerjee et al. (2025); Du et al. (2024); Xu et al. (2024); Zhao et al. (2024). Recently, the concept of a refusal feature, which encodes the refusal behavior of safety-aligned LLMs, is introduced, leveraging it in both adversarial attacks Arditi et al. (2024) and defense Yu et al. (2024). Building on the insight of the refusal feature, we further analyze the refusal feature and demonstrate its effectiveness in classifying prompts as harmful or harmless. Based on the capability of refusal feature, we propose a novel finetuning strategy for safe LLM finetuning.

**Defending Harmful Finetuning Attacks.** Harmful finetuning attacks are a subclass of jailbreaking techniques in which harmful input-output pairs are injected into the finetuning data, leading the model to generate unsafe outputs. The risks associated with harmful content in finetuning data have been highlighted in several studies (Lermen et al. (2023); Qi et al. (2023); Zhan et al. (2023); Hsu et al. (2024); He et al. (2024); Poppi et al. (2024); Betley et al. (2025); Hsiung et al. (2025); Xiao et al. (2025)). This makes preserving safety-alignment against harmful finetuning attacks increasingly critical, especially as AI providers begin offering FaaS. To address this issue, prior works proposed solutions targeting the alignment stage, the finetuning stage, or the post-finetuning stage. First, alignment-stage solutions aim to obtain robust safety-aligned LLM weights against harmful finetuning attacks, typically through regularization techniques based on expected perturbations (Huang et al. (2024c;d); Liu et al. (2024); Rosati et al. (2024); Tamirisa et al. (2024)). Second, finetuning-stage solutions preserve safety during finetuning on user data by freezing safety-critical parameters (Li et al. (2024a); Wei (2024); Li et al. (2025)) or incorporating safety regularization (Mukhoti et al. (2023); Huang et al. (2024b); Qi (2024); Yang et al. (2025)), often with additional safety-alignment data as guidance. Lastly, post-finetuning-stage solutions analyze differences between safety-aligned and finetuned models, and then edit model weights to compensate for safety degradation (Huang et al. (2024a); Hsu et al. (2024); Yi et al. (2025)).

Beyond these methods, recent works have examined how feature-space similarity. Hsiung et al. (2025) shows that safety guardrails weaken when downstream data representations overlap with safety-alignment data, while Xiao et al. (2025) reveals that benign prompt styles applied to harmful inputs can bypass safety mechanisms. Both findings relate to our refusal-feature-similarity–based filtering, which also addresses feature-level similarity between harmful and benign data.

In contrast to prior works following two-stage pipeline, we propose a Refusal-Teacher (Ref-Teacher)-guided finetuning framework, which directly finetunes the base model under the guidance of the Ref-Teacher, achieving better performance in both safety and downstream tasks.

## 3 Problem Setting

**Scenario.** In Finetuning-as-a-Service (FaaS), AI providers pursue two primary objectives: (i) achieving high user-specific task performance and (ii) preserving the safety-alignment of customized LLMs. To address these goals, we consider two distinct phases: the *alignment stage* (service preparation) and the *finetuning stage* (service provision). In the alignment stage, service providers are assumed to have access to a dataset of 5,000 harmful prompts and 5,000 harmless prompts, where each harmful prompt is paired with a refusal response. In the finetuning stage, users submit custom datasets to the provider for LLM customization. Importantly, providers have neither prior knowledge of whether user data contains harmful prompts nor its distribution during the alignment stage.

**Threat Models.** We assume that user data contains $p\%$ harmful prompts with harmful responses, while the remaining $(1-p)\%$ consists of harmless prompts sampled from the same dataset. When $p = 0$, the dataset includes only harmless prompts. Importantly, users do not inform which prompts are harmful or harmless, thereby exposing LLMs to the risk of safety degradation during finetuning. At the same time, LLMs are expected to achieve strong performance on user-specific downstream tasks while preserving their safety-alignment, making the problem particularly challenging.

## 4 Motivation: Safety-Aligned Weights are Not Enough.

Prior works on defending against harmful finetuning attacks have adopted a two-stage pipeline: first performing safety-alignment on an LLM, and then finetuning the safety-aligned model on user data. However, we find this paradigm suboptimal. After an LLM is safety-aligned, its weights are biased toward safety objectives, weakening initialization for downstream task learning compared to the base model. As a result, finetuning a safety-aligned model solely on user data yields limited task performance and degraded safety-alignment. In contrast, we observe that **directly finetuning the base model on both user data and safety-alignment data is more effective**. This strategy leverages the well-known fact that base models provide strong initialization for downstream tasks.

To validate this claim, we evaluate the transferability of safety-aligned models and base model by comparing two finetuning strategies via Harmful Score (HS) and Finetuning Accuracy (FA) after

Table 1: Performance comparison of various safety-aligned LLMs and base model finetuning under varying ratios $p$ of harmful prompts in user data. *SA* denotes safety-alignment and *FT* denotes finetuning. Numbers in $(\cdot)$ indicate the amount of data used for safety-alignment or finetuning.

| Methods | Harmful Score ($\downarrow$) | | | | Finetune Accuracy ($\uparrow$) | | | |
|---|---|---|---|---|---|---|---|---|
| | $p=0$ | $p=0.1$ | $p=0.3$ | $p=0.5$ | $p=0$ | $p=0.1$ | $p=0.3$ | $p=0.5$ |
| SA (1,000) $\rightarrow$ FT (1,000) | 4.9 | 48.1 | 78.2 | 79.8 | 42.8 | 43.4 | 40.2 | 42.7 |
| SA (5,000) $\rightarrow$ FT (1,000) | 3.3 | 22.8 | 61.7 | 71.1 | 41.3 | 41.9 | 39.4 | 39.7 |
| SA (10,000) $\rightarrow$ FT (1,000) | 2.2 | 16.2 | 57.3 | 71.3 | 41.1 | 39.9 | 39.1 | 37.1 |
| Repnoise (Rosati et al. (2024)) | 2.7 | 29.9 | 67.0 | 75.7 | 37.4 | 37.0 | 36.3 | 36.0 |
| Vaccine (Huang et al. (2024d)) | 1.3 | 5.4 | 35.0 | 57.5 | 22.9 | 23.2 | 21.7 | 20.3 |
| Booster (Huang et al. (2024c)) | 2.3 | 5.9 | 65.1 | 75.0 | 44.5 | 44.0 | 44.4 | 43.5 |
| Base $\rightarrow$ SA (1,000) + FT (1,000) | **0.9** | **2.0** | **4.3** | **15.7** | **47.6** | **47.9** | **45.6** | **45.0** |

Table 2: Gradient conflicts in two finetuning frameworks, measured by the cosine similarity between gradients from each objective during 300 finetuning steps. *SA* denotes safety alignment and *FT* denotes finetuning. Numbers in $(\cdot)$ indicate data size. *Freq* represents the frequency of conflicts, while *Avg* represents average cosine similarity. $p$ denotes the ratio of harmful prompts in user data.

| Methods | $p=0$ | | $p=0.1$ | | $p=0.3$ | | $p=0.5$ | |
|---|---|---|---|---|---|---|---|---|
| | Freq (%) | Avg | Freq (%) | Avg | Freq (%) | Avg | Freq (%) | Avg |
| SA (1,000) $\rightarrow$ FT (1,000) | 3.37 | 0.574 | 3.54 | 0.551 | 3.54 | 0.531 | 3.45 | 0.525 |
| SA (5,000) $\rightarrow$ FT (1,000) | 4.27 | 0.540 | 3.86 | 0.525 | 4.71 | 0.500 | 4.30 | 0.487 |
| SA (10,000) $\rightarrow$ FT (1,000) | 3.29 | 0.549 | 3.93 | 0.524 | 4.03 | 0.501 | 4.13 | 0.525 |
| Base $\rightarrow$ SA (1,000) + FT (1,000) | 35.09 | 0.110 | 36.80 | 0.099 | 40.80 | 0.073 | 46.03 | 0.039 |

finetuning (see Section 6 for metric details): (i) finetuning safety-aligned models solely on user data, and (ii) directly finetuning the base model on both user data and safety-alignment data. As shown in Table 1, stronger safety-aligned models preserve safety more effectively but exhibit weaker downstream task performance. In contrast, directly finetuning the base model achieves both robust safety-alignment and strong downstream task performance. In this strategy, safety-alignment data compensates the safety degradation caused by harmful finetuning attacks, while the base model's strong initialization supports effective downstream task learning. Remarkably, even this simple strategy achieves performance comparable to existing baselines in both safety and downstream task.

**Limitations.** However, directly finetuning the base model on both user data and safety-alignment data introduces **gradient conflicts**, as the model must simultaneously optimize two distinct objectives. Gradient conflict is defined as opposing update directions between gradients from different objectives, typically indicated by negative cosine similarity (Yu et al. (2020); Chen et al. (2020)). To quantify these conflicts, we measure cosine similarities between gradients from user data and safety-alignment data for each parameter, and record the cumulative frequency of negative similarities along with the average cosine similarity over 300 training steps (see Appendix A.3 for this choice). As shown in Table 2, when a safety-aligned model is finetuned only on user data, fewer than 5% of gradients conflict during training. In contrast, when the base model is finetuned on both user and safety-alignment data, more than 35% of gradients conflict, and the presence of harmful prompts in user data further exacerbates this issue. These gradient conflicts destabilize training.

Motivated by this observation, we propose a **Refusal-Teacher (Ref-Teacher)-based finetuning framework**, which alleviates gradient conflicts through alignment distillation and data filtering, thereby stabilizing training and enhancing robustness against harmful finetuning attacks.

## 5 METHOD: REFUSAL-TEACHER-GUIDED FINETUNING FRAMEWORK

We propose the **Refusal-Teacher (Ref-Teacher)-guided finetuning framework**, which directly finetunes the base model on both safety-alignment data and user data under the guidance of a Ref-Teacher via **alignment distillation** and **data filtering**. Unlike prior works that adopts the alignment stage, our approach introduces a **teacher preparation stage** to train the Ref-Teacher, followed by a finetuning stage where the unaligned base model is trained with Ref-Teacher guidance. An overview of our finetuning framework and a comparison with prior works are illustrated in Fig. 1.

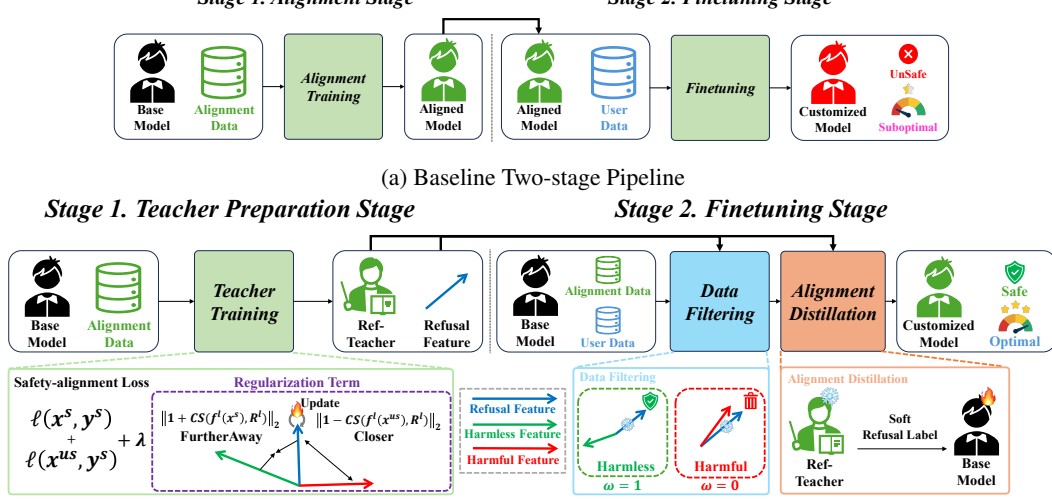

(a) Baseline Two-stage Pipeline

(b) Ref-Teacher-guided Finetuning Framework

Figure 1: Overview comparison of finetuning frameworks. (a) The base model is first trained on safety-alignment data and then finetuned on user data, which often results in safety degradation and limited downstream task performance. (b) Ref-Teacher is trained on safety-alignment data utilizing refusal feature, and then the base model is directly finetuned on both user data and safety-alignment data under the guidance of Ref-Teacher via data filtering and alignment distillation.

## 5.1 TEACHER PREPARATION STAGE

The goal of the teacher preparation stage is to train a safety-aligned teacher model for alignment distillation and data filtering during finetuning stage. To this end, we leverage the **refusal feature** during safety-alignment to train the model to accurately distinguish harmful from harmless prompts.

The refusal feature (Arditi et al. (2024)) is a one-dimensional representation that encodes safety behavior, namely refusing harmful prompts while generating helpful responses for harmless ones. Formally, it is defined as the mean difference between feature representations of harmful and harmless prompts at a specific layer $l$ of the LLM. Let $x^s$ and $x^{us}$ denote safe and unsafe prompts, respectively, and let $f^l(\cdot)$ denote the features of the last input token extracted from layer $l$. The refusal feature $R^l$ is computed as $R^l = \frac{1}{N_{us}} \sum_{i=1}^{N_{us}} f^l(x_i^{us}) - \frac{1}{N_s} \sum_{i=1}^{N_s} f^l(x_i^s)$ where $N_{us}$ and $N_s$ denote the number of unsafe and safe prompts, respectively. Consequently, the refusal feature exhibit high cosine similarity with harmful prompt features and low similarity with harmless prompt features, enabling harmful and harmless prompts classification via a cosine similarity threshold.

Leveraging this property, we develop the **Ref-Teacher**, a safety-aligned LLM that (i) generates soft refusal labels for alignment distillation and (ii) more effectively distinguishes harmful from harmless prompts using its refusal feature for data filtering. To achieve two objectives, we train the model with a safety-alignment loss, a supervised loss on safety-alignment data where harmful prompts are paired with refusal responses and harmless prompts with helpful outputs. This loss encourages the model to refuse harmful requests while producing appropriate responses to harmless ones, thereby enforcing distinct behaviors across different prompt types.

To further enhance discrimination, we introduce a **regularization term** that enforces clearer separation between harmful and harmless prompt features based on the refusal feature. Specifically, this term encourages the cosine similarity between a refusal feature and harmful prompt features to approach 1, while pushing the similarity with harmless prompt features toward $-1$. To prevent corruption of internal representations, we control its strength using a hyperparameter $\lambda$. The final objective for teacher preparation stage is consist of the safety-alignment loss and this regularization:

$$\mathcal{L}_{\text{teacher}} = \frac{1}{N} \sum_{i=1}^{N} \left[ \ell(x_i^s, y_i^s) + \ell(x_i^{us}, y_i^r) + \lambda \big\{ \|1 + \text{CS}(f^\ell(x_i^s), R^\ell)\|_2 + \|1 - \text{CS}(f^\ell(x_i^{us}), R^\ell)\|_2 \big\} \right] \quad (1)$$

where $\ell(\cdot, \cdot)$ denotes the cross-entropy loss, $CS(\cdot, \cdot)$ represents cosine similarity, $y^s$ and $y^r$ are the harmless and refusal responses, respectively, and $N$ is the number of training samples. As a result, the Ref-Teacher can generate appropriate refusal responses for harmful prompts while reliably distinguishing harmful from harmless inputs using its refusal feature.

In addition, we assume a setting where a pre-aligned model is unavailable, making it impossible to extract the refusal feature in advance. To address this, we dynamically update the refusal feature during training at fixed intervals (cycles) based on its definition. For each training step, harmful and harmless prompts are accumulated into sets $S_{us}$ and $S_s$, and the refusal feature is updated for every cycle. Before the first update, we set $\lambda = 0$ to disable regularization, as the refusal feature is not yet reliable. This **dynamic update strategy**

---

**Algorithm 1** Training Process of the Ref-Teacher Model

**Require:** Unsafe data $x^{us}$, Safe data $x^s$, Cycle number $C$, LoRA weight $W$, Regularization strength $\lambda$, Learning rate $\eta$
**Ensure:** Trained LoRA weight $W$, Refusal Feature $R^l$
    Initialize Unsafe prompt set $S_{us} \leftarrow []$
    Initialize Safe prompt set $S_s \leftarrow []$
    Initialize Refusal feature $R^l \leftarrow None$
    Initialize Counter $c \leftarrow 0$
    **while** not converged **do**
        Sample $B$ examples each of $x^{us}$ and $x^s$
        Append $x^{us}$ to $S_{us}$
        Append $x^s$ to $S_s$
        $c \leftarrow c + B$
        **if** $c \geq C$ **then**
            Update $R^l \leftarrow \frac{1}{|S_{us}|} \sum_{x \in S_{us}} f^l(x) - \frac{1}{|S_s|} \sum_{x \in S_s} f^l(x)$
            Reset Unsafe prompt set $S_{us} \leftarrow []$
            Reset Safe prompt set $S_s \leftarrow []$
            $c \leftarrow 0$
        **end if**
        **if** $R^l$ is None **then**
            $\lambda \leftarrow 0$
        **end if**
        Compute $\mathcal{L}_{teacher}$ from Eq. 1
        Update $W \leftarrow W - \eta \cdot \nabla \mathcal{L}_{teacher}$
    **end while**
    **return** $W$ and $R^l$

---

removes the need for a separate alignment stage, enabling the model to compute refusal feature and learn discriminative representations within a single training process. The complete algorithm for the teacher preparation stage is provided in Alg. 1.

## 5.2 FINETUNING STAGE

In the finetuning stage, the Ref-Teacher is frozen and serves as a teacher for two complementary purposes: (i) providing alignment distillation and (ii) filtering harmful prompts from user data. This approach enables the base model to effectively learn user-specific tasks while maintaining strong safety-alignment by mitigating gradient conflicts that arise during finetuning.

**Alignment Distillation.** Knowledge distillation is a widely used technique for mitigating gradient conflicts in multi-objective learning. Prior works (Hinton et al. (2015); Furlanello et al. (2018); Müller et al. (2019); Yuan et al. (2020)) show that soft labels from a teacher provide richer supervision and yield smoother loss surfaces than hard labels. Following this principle, we adopt alignment distillation to guide the base model when learning both user-specific tasks and safety-alignment. Specifically, the Ref-Teacher generates soft refusal labels, and the base model is trained with (i) a supervised loss on user data and (ii) a KL-divergence loss on safety-alignment data to align its predictions with the Ref-Teacher's soft labels. This distillation stabilizes training by reducing gradient conflicts, resulting in safe and appropriate responses for both harmful and user-specific inputs.

To ensure the reliability of these soft refusal labels, we reuse the safety-alignment data from the teacher preparation stage. Since the Ref-Teacher has already been trained on this data, it can generate accurate refusal responses. Moreover, as shown in Table 1, only a small subset of this data is needed to be reused, removing the need for additional alignment data for finetuning stage.

**Data Filtering.** While alignment distillation mitigates gradient conflicts between safety and user-specific task objectives, it alone cannot prevent these conflicts from being exacerbated by harmful finetuning attacks. To address this, we adopt data filtering as a complementary solution. In our framework, the Ref-Teacher filters harmful prompts from user data by leveraging its refusal feature to distinguish harmful from harmless inputs. Specifically, harmful data are identified by measuring the cosine similarity between the refusal feature $R^l$ and the feature $f^l(x_i)$ of each input prompt. If the similarity exceeds a predefined threshold $\tau$, the prompt is classified as harmful, otherwise as

harmless. This filtering mechanism is formulated as a binary filtering indicator $\omega_i$:

$$\omega_i = \begin{cases} 0, & \text{if} \quad CS(R^l, f^l(x_i)) > \tau \\ 1, & otherwise \end{cases}. \tag{2}$$

In Eq. 2, prompts classified as harmful are excluded from the supervised finetuning loss by setting $\omega_i = 0$, since misclassifying harmful prompts as harmless could exacerbate gradient conflicts and destabilize training. To improve recall in harmful prompt classification, we set the threshold relatively high, ensuring that the Ref-Teacher is less likely to misclassify harmful prompts as harmless (even at the cost of discarding some harmless ones). Consequently, all data predicted to be harmful are discarded, ensuring finetuning is performed only on harmless prompts. This strategy preserves safety and stabilizes training by preventing even small amounts of harmful data.

**Overall Objective.** Our Ref-Teacher-guided finetuning strategy incorporates the dual-teacher mechanism, combining supervised finetuning on user data with alignment distillation on safety-alignment data. The overall loss function for finetuning stage is defined as:

$$\mathcal{L}_{ft} = \frac{1}{N_{user}} \sum_{i=1}^{N_{user}} \omega_i * \ell(x_i, y_i) + \alpha T^2 * \frac{1}{N_{align}} \sum_{i=1}^{N_{align}} \text{KL}(p_{t,i}^T \,||\, p_{s,i}^T), \tag{3}$$

where $\ell(x_i, y_i)$ is the cross-entropy loss on user data $(x_i, y_i)$ weighted by $\omega_i$. The second term is the alignment distillation loss on safety-alignment data, where KL denotes KL-divergence between the teacher (Ref-Teacher) distribution $p_{t,i}^T$ and the student (base model) distribution $p_{s,i}^T$ at temperature $T$. The softened distribution is $p_i^T = \frac{\exp(z_i/T)}{\sum_{j=1}^V \exp(z_j/T)}$ where $z$ denotes the model logits and $V$ is the vocabulary size. The hyperparameter $\alpha$ controls the relative weight of the distillation term.

## 6 EXPERIMENT

We evaluate the effectiveness of our finetuning framework on safety-alignment and user-specific task performance under various settings. We varied the ratio of harmful prompts, the size of user data, the type of harmless prompts (GSM8K (Cobbe et al. (2021)), SST2 (Socher et al. (2013)), AGNEWS (Zhang et al. (2015)), AlpacaEval (Li et al. (2023))), and the base model (Llama3-8B (Llama Team (2024)), Gemma2-9B (Team et al. (2024)), Qwen2-7B (Team (2024))). Unless noted otherwise, we used Llama3-8B, 0.1 poison ratio, 1,000 user data, and GSM8K as harmless data.

**Datasets.** For teacher preparation stage, we used $N = 5,000$ harmful prompts with refusal responses from BeaverTails (Ji et al. (2023)), and $N = 5,000$ harmless prompts with helpful responses from Alpaca (Taori et al. (2023)). For finetuning stage, user data was constructed by mixing harmful and harmless samples with a specific poison ratio. The alignment data size $N_{align}$ was set equal to the user data size $N_{user}$. All harmful prompts in experiments were sourced from BeaverTails, but distinct subsets were used for the teacher preparation, finetuning, and evaluation to avoid overlap.

**Metrics.** We evaluate both safety-alignment and task performance using two metrics: Harmful Score (HS) and Finetuning Accuracy (FA), following prior works (Huang et al. (2024a;b;c;d; 2025); Liu et al. (2025)). HS is the proportion of harmful responses among 1,000 outputs generated from BeaverTails test set, classified by the pretrained moderation model Beaver-Dam-7B (Ji et al. (2023)). FA is measured by downstream benchmarks for GSM8K, SST2, AGNEWS, and AlpacaEval, using 1,000, 872, 1,000, and 122 samples, respectively. AlpacaEval was assessed by GPT-4o (Hurst et al. (2024)), following standard practices. Both HS and FA were evaluated after finetuning stage.

**Baselines.** We compare our framework against both alignment and finetuning-stage solutions. **SFT** is the standard supervised learning, aligning on harmful prompt-refusal pairs and then finetuning on user data. Among alignment-stage methods, **RepNoise** (Rosati et al. (2024)) removes harmful representations, **Vaccine** (Huang et al. (2024d)) enforces embedding invariance via perturbations, and **Booster** (Huang et al. (2024c)) simulates harmful finetuning to regularize harmful loss. All are followed by finetuning the aligned model on user data. For finetuning-stage solutions, applied to SFT-aligned models, **LDIFS** (Mukhoti et al. (2023)) constrains concept forgetting, while **Lisa** (Huang et al. (2024b)) alternates optimization between alignment and user data with a regularization term.

Table 3: Performance under varying harmful prompts ratios $p$ in user data. Lower harmful scores ($\downarrow$) and higher finetuning accuracy ($\uparrow$) indicate better performance. Results are averaged over seeds 30, 42, and 50. Finetuning accuracy is not reported for $p = 1.0$ since harmless data is unavailable.

| Methods | Harmful Score ($\downarrow$) | | | | | Finetune Accuracy ($\uparrow$) | | | | |
|---|---|---|---|---|---|---|---|---|---|---|
| | $p=0$ | $p=0.1$ | $p=0.3$ | $p=0.5$ | $p=1.0$ | $p=0$ | $p=0.1$ | $p=0.3$ | $p=0.5$ | $p=1.0$ |
| SFT | $2.2_{\pm0.1}$ | $16.2_{\pm0.4}$ | $57.3_{\pm0.6}$ | $71.3_{\pm0.6}$ | $76.7_{\pm0.4}$ | $41.1_{\pm0.0}$ | $39.9_{\pm0.6}$ | $39.1_{\pm0.2}$ | $37.1_{\pm0.6}$ | - |
| Repnoise (Rosati et al. (2024)) | $2.7_{\pm0.4}$ | $29.9_{\pm0.6}$ | $67.0_{\pm5.1}$ | $75.7_{\pm3.1}$ | $79.7_{\pm0.6}$ | $37.4_{\pm0.3}$ | $37.0_{\pm1.2}$ | $36.3_{\pm0.7}$ | $36.0_{\pm1.4}$ | - |
| Vaccine (Huang et al. (2024d)) | $1.3_{\pm0.2}$ | $5.4_{\pm0.7}$ | $35.0_{\pm0.3}$ | $57.5_{\pm0.4}$ | $81.3_{\pm0.1}$ | $22.9_{\pm0.5}$ | $23.2_{\pm1.0}$ | $21.7_{\pm0.3}$ | $20.3_{\pm0.4}$ | - |
| Booster (Huang et al. (2024c)) | $2.3_{\pm0.1}$ | $5.9_{\pm0.2}$ | $65.1_{\pm0.3}$ | $75.0_{\pm0.6}$ | $79.0_{\pm0.4}$ | $44.5_{\pm0.6}$ | $44.0_{\pm0.9}$ | $44.4_{\pm0.6}$ | $43.5_{\pm0.6}$ | - |
| LDIFS (Mukhoti et al. (2023)) | $1.0_{\pm0.2}$ | $4.1_{\pm0.7}$ | $7.1_{\pm0.2}$ | $14.7_{\pm0.3}$ | $24.0_{\pm0.4}$ | $18.0_{\pm0.9}$ | $16.7_{\pm0.8}$ | $15.5_{\pm0.1}$ | $15.4_{\pm0.6}$ | - |
| Lisa (Huang et al. (2024b)) | $1.4_{\pm0.2}$ | $5.3_{\pm0.1}$ | $25.9_{\pm1.5}$ | $49.2_{\pm0.7}$ | $67.3_{\pm1.0}$ | $38.3_{\pm0.7}$ | $38.9_{\pm0.9}$ | $37.8_{\pm0.9}$ | $36.2_{\pm0.5}$ | - |
| Ref-Teacher (Ours) | $\mathbf{0.9}_{\pm0.3}$ | $\mathbf{1.0}_{\pm0.5}$ | $\mathbf{0.6}_{\pm0.1}$ | $\mathbf{0.9}_{\pm0.3}$ | $\mathbf{1.3}_{\pm0.2}$ | $\mathbf{48.8}_{\pm0.5}$ | $\mathbf{49.0}_{\pm0.5}$ | $\mathbf{45.5}_{\pm0.9}$ | $\mathbf{44.8}_{\pm0.5}$ | - |

Table 4: Performance comparison across varying amounts of user data. $n$ denotes the user data size.

| Methods | Harmful Score ($\downarrow$) | | | | | Finetune Accuracy ($\uparrow$) | | | | |
|---|---|---|---|---|---|---|---|---|---|---|
| | n=1000 | n=1500 | n=2000 | n=2500 | Average | n=1000 | n=1500 | n=2000 | n=2500 | Average |
| SFT | 16.7 | 39.4 | 55.8 | 63.9 | 44.0 | 40.6 | 42.9 | 44.5 | 45.3 | 43.3 |
| Repnoise (Rosati et al. (2024)) | 30.4 | 50.4 | 61.7 | 72.9 | 53.9 | 38.4 | 40.5 | 43.6 | 43.5 | 41.5 |
| Vaccine (Huang et al. (2024d)) | 4.8 | 19.8 | 34.1 | 45.0 | 25.9 | 24.4 | 28.5 | 31.3 | 33.9 | 29.5 |
| Booster (Huang et al. (2024c)) | 5.9 | 19.4 | 48.2 | 62.6 | 34.0 | 43.4 | 45.3 | 48.4 | 48.5 | 46.4 |
| LDIFS (Mukhoti et al. (2023)) | 4.0 | 5.7 | 4.7 | 6.0 | 5.1 | 17.0 | 16.7 | 17.7 | 18.4 | 17.5 |
| Lisa (Huang et al. (2024b)) | 5.3 | 8.2 | 10.4 | 12.8 | 9.2 | 38.3 | 37.8 | 40.3 | 42.7 | 39.8 |
| Ref-Teacher (Ours) | **0.5** | **0.9** | **0.9** | **1.0** | **0.8** | **49.0** | **50.1** | **52.1** | **51.8** | **50.8** |

## 6.1 EXPERIMENT RESULTS

**Robustness under Varying Harmful Prompt Ratio.** We evaluate our framework using HS and FA under varying ratios of harmful prompts $p$ in user data, ranging from fully clean data ($p = 0$) to entirely harmful data ($p = 1.0$). Table 3 shows that our method consistently achieves the lowest HS and the highest FA across all values of $p$, outperforming all baselines. This effectiveness and robustness stems from directly finetuning the base model while mitigating gradient conflicts under harmful finetuning attacks through alignment distillation and data filtering with the Ref-Teacher. Moreover, alignment-stage baselines such as RepNoise (Rosati et al. (2024)), Vaccine (Huang et al. (2024d)), and Booster (Huang et al. (2024c)) degrade under high harmful ratios ($p \geq 0.3$), while finetuning-stage solutions such as LDIFS (Mukhoti et al. (2023)), Lisa (Huang et al. (2024b)), and our approach remain robust, maintaining lower HS. Among these, our Ref-Teacher-guided finetuning framework achieves the best performance in both safety-alignment and user-specific downstream tasks.

**Scalability with Varying Amounts of User Data.** We evaluate scalability of our framework by measuring HS and FA as the number of user data samples increases from 1,000 to 2,500. As shown in Table 4, our Ref-Teacher–guided finetuning strategy consistently achieves the best performance across all settings. For a fixed poison ratio, our method maintains low HS even as the absolute number of harmful prompts grows with data size, demonstrating strong robustness in safety-alignment. At the same time, FA improves as more user data become available for user-specific tasks. These results validate the scalability and adaptability of our approach across varying data scales.

**Generalization across Diverse Finetuning Datasets.** In our default setting, GSM8K serves as the user-specific downstream task. To evaluate generalization across datasets, we replaced the harmless portion of user data with SST2, AGNEWS, and AlpacaEval samples, and measured HS and FA for our method and baselines. As shown in Table 5, our approach consistently yields the lowest HS and highest FA across all datasets. These results demonstrate the strong generalization of our framework, preserving both safety-alignment and task performance across diverse downstream tasks.

**Adaptability across Model Architectures.** We assess adaptability to diverse model architectures by training the Ref-Teacher on Gemma2-9B and Qwen2-7B, and finetuning each corresponding base model on safety-alignment and user data. To obtain the refusal feature, we select the optimal safety layer for harmfulness classification, which differs by architecture (Details are in Appendix B.1). Table 6 shows that our method consistently reduces harmfulness while improving user-specific downstream performance across model architectures. These results demonstrate that our approach generalizes across diverse LLM backbones rather than being restricted to a single architecture.

Table 5: Performance comparison across different downstream tasks.

| Methods | GSM8K | | SST2 | | AGNEWS | | AlpacaEval | | Average | |
|---|---|---|---|---|---|---|---|---|---|---|
| | HS ↓ | FA ↑ | HS ↓ | FA ↑ | HS ↓ | FA ↑ | HS ↓ | FA ↑ | HS ↓ | FA ↑ |
| SFT | 16.7 | 40.6 | 33.5 | 93.4 | 28.2 | 82.8 | 23.7 | 32.7 | 20.4 | 49.9 |
| Repnoise (Rosati et al. (2024)) | 30.4 | 38.4 | 63.0 | 93.4 | 58.6 | 84.6 | 45.4 | 29.3 | 39.5 | 49.1 |
| Vaccine (Huang et al. (2024d)) | 4.8 | 24.4 | 35.8 | 90.0 | 29.5 | 83.2 | 55.8 | 14.4 | 25.2 | 42.4 |
| Booster (Huang et al. (2024c)) | 5.9 | 43.4 | 9.2 | 93.6 | 5.3 | 85.3 | 29.4 | 34.0 | 10.0 | 51.3 |
| LDIFS (Mukhoti et al. (2023)) | 4.0 | 17.0 | 14.6 | 90.5 | 12.5 | 71.2 | 5.7 | 33.7 | 7.4 | 42.5 |
| Lisa (Huang et al. (2024b)) | 5.3 | 38.3 | 21.4 | 93.4 | 14.9 | 84.5 | 10.1 | 29.6 | 10.3 | 49.2 |
| Ref-Teacher (Ours) | **0.5** | **49.0** | **1.3** | **94.5** | **1.2** | **86.1** | **2.4** | **34.6** | **1.1** | **52.8** |

Table 6: Performance comparison across different model architectures. Our Ref-Teacher-guided finetuning strategy shows strong adaptability across Llama3-8B, Gemma2-9B, and Qwen2-7B.

| Methods | Llama3-8B | | Gemma2-9B | | Qwen2-7B | | Average | |
|---|---|---|---|---|---|---|---|---|
| | HS ↓ | FA ↑ | HS ↓ | FA ↑ | HS ↓ | FA ↑ | HS ↓ | FA ↑ |
| SFT | 16.7 | 40.6 | 26.4 | 59.5 | 37.9 | 66.8 | 27.0 | 55.6 |
| Repnoise (Rosati et al. (2024)) | 30.4 | 38.4 | 26.2 | 57.1 | 25.4 | 63.7 | 27.3 | 53.1 |
| Vaccine (Huang et al. (2024d)) | 4.8 | 24.4 | 18.0 | 52.5 | 10.2 | 63.6 | 11.0 | 46.8 |
| Booster (Huang et al. (2024c)) | 5.9 | 43.4 | 2.3 | 58.4 | 4.9 | **70.0** | 4.4 | 57.3 |
| LDIFS (Mukhoti et al. (2023)) | 4.0 | 17.0 | 3.1 | 36.0 | 10.7 | 64.1 | 5.9 | 39.0 |
| Lisa (Huang et al. (2024b)) | 5.3 | 38.3 | 6.2 | 54.5 | 4.4 | 61.6 | 5.3 | 51.5 |
| Ref-Teacher (Ours) | **0.5** | **49.0** | **1.3** | **63.6** | **0.6** | 69.7 | **0.8** | **60.8** |

Table 7: Classification accuracy (%) during finetuning.

| Datasets | Harmful | Harmless | Total |
|---|---|---|---|
| GSM8K | 100.00 | 97.70 | 97.93 |
| SST2 | 99.91 | 95.30 | 95.76 |
| AGNEWS | 99.91 | 99.86 | 99.87 |
| AlpacaEval | 99.90 | 77.04 | 79.33 |

Table 8: F1 Scores (%) of Ref-Teacher, guardrail models, and linear classifier across various jailbreaking attacks.

| Datasets | BeaverTails | JailbreakBench | Toxic-chat | GCG | AutoDAN-turbo |
|---|---|---|---|---|---|
| Linear Classifier | 83.5 | 69.8 | 75.7 | 52.4 | 48.4 |
| LLaMAGuard3-8B | 64.1 | **88.7** | 57.0 | 89.7 | 9.3 |
| OpenAI Moderation | 67.8 | 74.7 | 44.4 | 81.0 | 52.2 |
| Ref-Teacher ($\tau = 0$) | **93.4** | 79.8 | **87.0** | **92.9** | **82.1** |

Table 9: Ablation study on safety and task performance.

| AD | Filtering | HS ↓ | FA ↑ |
|---|---|---|---|
| X | X | 2.0 | 47.9 |
| O | X | 2.2 | 46.2 |
| X | O | 0.6 | 46.5 |
| O | O | 0.5 | 49.0 |

Table 10: Ablation study on gradient conflicts.

| AD | Filtering | $p = 0$ | | p=0.1 | | p=0.3 | | p=0.5 | |
|---|---|---|---|---|---|---|---|---|---|
| | | Freq (%) | Avg | Freq (%) | Avg | Freq (%) | Avg | Freq (%) | Avg |
| X | X | 35.09 | 0.110 | 36.80 | 0.099 | 40.80 | 0.073 | 46.03 | 0.039 |
| O | X | 32.26 | 0.131 | 34.02 | 0.117 | 37.78 | 0.090 | 42.55 | 0.055 |
| X | O | 36.11 | 0.102 | 36.51 | 0.097 | 37.80 | 0.087 | 39.91 | 0.073 |
| O | O | 30.02 | 0.140 | 29.60 | 0.143 | 28.93 | 0.145 | 28.29 | 0.149 |

## 6.2 ANALYSIS

**Classification Performance of Ref-Teacher.** We evaluate Ref-Teacher's ability to classify harmful and harmless prompts during finetuning on GSM8K, SST2, AGNEWS, and AlpacaEval, achieving near-perfect accuracy on harmful prompts and consistently high accuracy on harmless ones (Table 7). For generalization, we test on JailbreakBench harmless prompts combined with harmful prompts from BeaverTails, JailbreakBench, Toxic-chat, GCG, and AutoDAN-turbo. Ref-Teacher, trained only on BeaverTails (harmful) and Alpaca (harmless), is compared against LLaMAGuard3-8B (Llama Team (2024)), OpenAI Moderation, and a linear classifier trained on LLaMA3-8B features using the same data. As shown in Table 8, the classifier performs well on in-distribution but degrades on unseen jailbreaks, whereas Ref-Teacher consistently outperforms all baselines, achieving high F1 scores even on advanced attacks (GCG, AutoDAN-turbo). These results demonstrate the accuracy and generalization of refusal-based classification for reliable harmful data filtering.

**Ablation Study on Safety and Task Performance.** We assess the impact of alignment distillation (AD) and data filtering (Filtering) on safety and task performance by removing each component. As shown in Table 9, AD alone improves neither safety nor finetuning accuracy, indicating that it cannot stabilize optimization when harmful prompts remain in user data. In contrast, Filtering alone reduces harmfulness but lowers finetuning accuracy due to reduced user data, which increases overfitting risk. These results highlight their complementary roles: AD stabilizes optimization but requires filtered data, whereas Filtering reduces harmfulness but risks overfitting without distillation. Their combination synergistically achieves strong task performance while preserving safety alignment.

Table 11: Computational Overhead across Baselines.

| Methods | Alignment Stage | | Finetuning Stage | | Sum | |
|---|---|---|---|---|---|---|
| | GPUTime (s) | GPUMemory (GB) | GPUTime (s) | GPUMemory (GB) | GPUTime (s) | GPUMemory (GB) |
| SFT | 0.91 | 7.84 | 1.59 | 9.31 | 2.50 | 17.15 |
| Repnoise (Rosati et al. (2024)) | 4.27 | 15.27 | 1.59 | 9.31 | 5.86 | 24.58 |
| Vaccine (Huang et al. (2024d)) | 1.79 | 7.84 | 1.59 | 9.31 | 3.38 | 17.15 |
| Booster (Huang et al. (2024c)) | 3.92 | 9.39 | 1.59 | 9.31 | 5.51 | 18.70 |
| LDIFS (Mukhoti et al. (2023)) | 0.91 | 7.84 | 1.95 | 16.51 | 2.86 | 24.35 |
| Lisa (Huang et al. (2024b)) | 0.91 | 7.84 | 1.59 | 9.02 | 2.50 | 16.86 |
| Ref-Teacher (Ours) | 1.76 | 11.29 | 1.84 | 12.01 | 3.60 | 23.30 |

**Ablation Study on Gradient Conflicts.** We evaluate the contributions of alignment distillation (AD) and data filtering (Filtering) on gradient conflicts by removing each component and varying the harmful ratio $p$. Table 10 shows that AD alone reduces conflicted parameters on clean data but loses effectiveness as $p$ increases, while Filtering alone stabilizes the frequency of conflicts but does not sufficiently mitigate it. Consequently, AD and Filtering complement each other in our framework, mitigating gradient conflicts effectively under harmful finetuning attacks.

**Computational Overhead.** To quantify the computational overhead introduced by the Teacher Preparation Stage, we measured both GPUTime and GPUMemory for the alignment stage and the finetuning stage separately. All measurements were performed on four RTX 3090 GPUs, and Table X reports the per-GPU GPUTime (average per-step runtime) and GPUMemory (average per-step memory usage). As shown in Table 11, while Ref-Teacher does incur additional cost relative to SFT, the increase is moderate compared to other baselines. Specifically, compared to SFT, Ref-Teacher uses 44.0% more GPUTime and 35.9% more GPUMemory, yet achieves a 93.8% reduction in harmful score and a 22.8% increase in finetuning accuracy. These results indicate that the computational overhead is modest and well-justified given the substantial safety and utility improvements.

## 7 CONCLUSION

In this work, we address a key limitation of current two-stage Finetuning-as-a-Service (FaaS) practices, where providers first safety-align an LLM and then finetune the safety-aligned model on user data. We observe that safety-aligned models offer weak initialization for downstream task learning, leading to suboptimal task performance and degraded safety when finetuning the safety-aligned model on user data. To overcome this, we introduce the Refusal-Teacher (Ref-Teacher)-guided finetuning framework, which directly finetunes the unaligned base model on both safety-alignment data and user data under the guidance of a safety-aligned Ref-Teacher via alignment distillation and data filtering. Extensive experiments demonstrate that our framework consistently achieves the lowest harmful scores and the highest finetuning accuracy across diverse settings, outperforming baselines. Overall, our approach offers a practical and effective solution for FaaS, ensuring strong user-specific task performance while preserving safety-alignment against harmful finetuning attacks.

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

# APPENDIX

## A EXPERIMENT DETAILS

### A.1 TRAINING SETUP

In the teacher preparation stage, we train the Refusal-Teacher (Ref-Teacher) model for 20 epochs using batches of size 10, consisting of 5 harmful and 5 harmless prompts, with a learning rate of $5e^{-4}$. During the finetuning stage, we train the base model with Ref-Teacher for 20 epochs using 20 batches (10 harmful data and 10 harmless data), with a learning rate of $1e^{-5}$. For the AlpacaEval dataset (Li et al. (2023)), due to its small size, we train the base model for 100 epochs using 700 prompts. In both stages, we apply LoRA (Hu et al. (2022)) with a rank of 32, targeting the query, key, and value components of the attention modules. Also, we use the AdamW optimizer (Loshchilov & Hutter (2017)) with a weight decay of 0.1 and a constant learning rate schedule. All experiments are conducted on four RTX3090 GPUs.

### A.2 HYPERPARAMETERS FOR OUR METHOD

Our proposed framework introduces several additional hyperparameters. First, in teacher preparation stage, we set the regularization strength for training Ref-Teacher model to $\lambda = 0.1$. Refusal features are extracted from specific layer in LLMs: $l = 12$ for LLAMA3-8B, $l = 11$ for Gemma2-9B, $l = 18$ for Qwen2-7B. The refusal features are updated periodically every $C = 6$ cycles, with each update performed using 30 harmful and 30 harmless prompts. During finetuning stage, for harmful and harmless classification using the Ref-Teacher model, we use a threshold of 0.9 to maximize the recall of harmful prompts. For alignment distillation, we set the distillation strength $\alpha = 0.1$ and use a the temperature $T = 1$. Ablation studies to identify the optimal values for these hyperparameters are presented in Sec. B. All the other hyperparameters for the baseline methods follow the settings specified in their respective original papers (Mukhoti et al. (2023); Huang et al. (2024c;d;b); Rosati et al. (2024)).

### A.3 MEASURING GRADIENT CONFLICTS

We showed that directly finetuning the base model on both user data and safety-alignment data introduces gradient conflicts, which we measured using negative cosine similarities between gradients from the two datasets. Specifically, we reported the average frequency of negative cosine similarities and the average cosine similarity values accumulated over the first 300 training steps. We focus on this range because, after 300 steps, even when training on the same dataset, the signal-to-noise ratio (SNR) decreases sharply, making noise more dominant and causing negative cosine similarities to occur more frequently. Figure A2 reports the measured SNR when finetuning a safety-aligned model on user data, showing that SNR drops to very low values beyond 300 steps. Although gradients from the same dataset are theoretically expected to exhibit very few negative cosine similarities, we observed that their frequency increases after 300 steps under this finetuning setup. For this reason, we present negative cosine similarity statistics only up to 300 steps, as shown in Tables 2 and 10.

## B EXPERIMENTS FOR FINDING OPTIMAL HYPERPARAMETERS

### B.1 LAYER SELECTION FOR REFUSAL FEATURE EXTRACTION

The refusal feature reflects the model's ability to distinguish between harmful and harmless prompts and to generate refusal responses only for harmful inputs. Therefore, it is most effective to extract the refusal feature from a layer that maximizes the distinction between harmful and harmless prompt representations. Based on a prior work (Li et al. (2024a)) suggesting that such layers are typically located in the middle layers of LLMs, we identify the optimal layer by evaluating classification accuracy and the norm difference between the average features of harmful and harmless prompts across 8 different layers. As shown in Table A1, both the classification accuracy and norm differences vary across layers. For each layer, the classification threshold is optimized to maximize classification performance. As a result, we used $l = 11$ for the Gemma2-9B (Team et al. (2024)) and $l = 18$

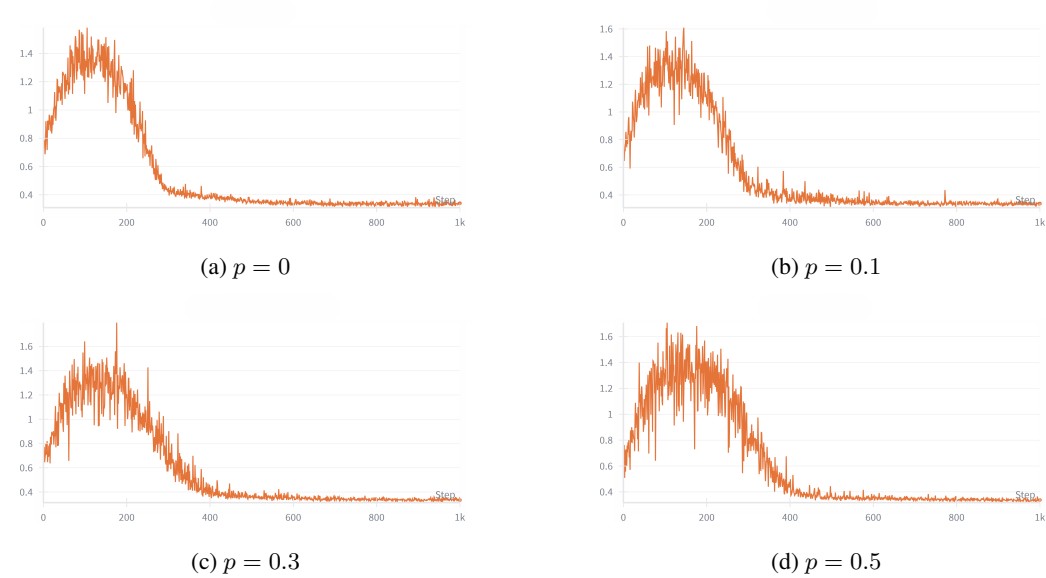

(a) $p = 0$      (b) $p = 0.1$

(c) $p = 0.3$      (d) $p = 0.5$

Figure A2: Signal-to-noise ratio (SNR) measured when finetuning a safety-aligned model solely on user data. SNR values consistently drop after 300 training steps across varying harmful ratios $p$, making noise dominant and increasing the frequency of negative cosine similarities between gradients.

Table A1: Classification accuracy and feature L1-norm differences across layers for identifying the optimal layer index used to extract refusal features in Gemma2-9B-it and Qwen2-7B-Instruct. The selected layer used in our experiments is highlighted in bold. For each layer, features are extracted from the last input token, and classification thresholds are optimized.

(a) Gemma2-9B-it

| Layer idx | Threshold | Harmful Acc (%) | Harmless Acc (%) | Acc (%) | Harmful Avg | Harmless Avg | Diff |
|---|---|---|---|---|---|---|---|
| 7 | 0.0055 | 76.6 | 93.4 | 85.0 | 0.0239 | -0.0090 | 0.0329 |
| 8 | 0.0225 | 69.8 | 93.8 | 81.8 | 0.0374 | 0.0080 | 0.0294 |
| 9 | 0.0510 | 89.6 | 96.6 | 93.1 | 0.0878 | 0.0303 | 0.0575 |
| 10 | 0.0530 | 93.8 | 95.0 | 94.4 | 0.0949 | 0.0363 | 0.0586 |
| **11** | **0.0245** | **96.2** | **98.6** | **97.4** | **0.0844** | **-0.0020** | **0.0864** |
| 12 | 0.0555 | 91.4 | 96.4 | 93.9 | 0.1133 | 0.0319 | 0.0814 |
| 13 | 0.0570 | 90.8 | 92.8 | 91.8 | 0.1285 | 0.0346 | 0.0939 |
| 14 | 0.184 | 86.6 | 91.2 | 88.9 | 0.2629 | 0.1524 | 0.0111 |

(b) Qwen2-7B-Instruct

| Layer idx | Threshold | Harmful Acc (%) | Harmless Acc (%) | Acc (%) | Harmful Avg | Harmless Avg | Diff |
|---|---|---|---|---|---|---|---|
| 13 | 0.046 | 96.4 | 98.6 | 97.5 | 0.1814 | 0.0153 | 0.1661 |
| 14 | 0.118 | 97.2 | 97.8 | 97.5 | 0.2622 | 0.0875 | 0.1747 |
| 15 | 0.060 | 98.0 | 98.2 | 98.1 | 0.2297 | 0.0265 | 0.2032 |
| 16 | 0.145 | 96.2 | 99.2 | 97.7 | 0.3003 | 0.1093 | 0.1910 |
| 17 | 0.164 | 98.6 | 97.8 | 98.2 | 0.3709 | 0.1326 | 0.2383 |
| **18** | **0.195** | **98.6** | **99.8** | **99.2** | **0.4166** | **0.1551** | **0.2615** |
| 19 | 0.163 | 97.4 | 99.6 | 98.5 | 0.3555 | 0.1262 | 0.2293 |
| 20 | 0.055 | 95.0 | 99.4 | 97.2 | 0.2458 | 0.0211 | 0.2247 |

for the Qwen2-7B (Team (2024)) in all of our experiments. For Llama3-8B, we adopted $l = 12$, following a prior work (Arditi et al. (2024)). Additionally, we used the feature corresponding to the last input token, as it encodes the entire sentence due to the language model's causal structure and attention masking.

Table A2: Effect of cycle (C) on the Ref-Teacher performance.

| Cycle | $N_{us} = N_s$ | HS ($\downarrow$) | FA ($\uparrow$) |
|---|---|---|---|
| **6** | **30** | **0.5** | **49.0** |
| 20 | 100 | 1.1 | 47.8 |
| 100 | 500 | 1.1 | 47.7 |
| 200 | 1000 | 1.2 | 46.8 |

Table A3: Varying $\lambda$.

| $\lambda$ | HS ($\downarrow$) | FA ($\uparrow$) |
|---|---|---|
| 0.05 | 0.7 | 48.4 |
| **0.1** | **0.5** | **49.0** |
| 0.3 | 1.0 | 48.3 |
| 0.5 | 1.0 | 48.3 |
| 1.0 | 1.6 | 47.7 |

Table A4: Varying Threshold.

| Threshold | HS ($\downarrow$) | FA ($\uparrow$) |
|---|---|---|
| 0 | 0.9 | 47.8 |
| 0.3 | 0.6 | 46.2 |
| 0.5 | 1.4 | 47.2 |
| 0.7 | 1.0 | 47.1 |
| **0.9** | **0.5** | **49.0** |

## B.2 EFFECT OF CYCLE LENGTH ON REFUSAL FEATURE UPDATES

During the teacher preparation stage, the cycle determines both the interval and the number of samples used to update the refusal feature, which serves as important reference for distinguishing between features of harmful and harmless prompts in our Ref-Teacher model. A short cycle updates the refusal feature more frequently but with fewer samples, which can lead to unstable training due to variance of refusal features. In contrast, a long cycle uses more samples for each update but, due to its infrequent updates, may overfit to suboptimal refusal feature. Table A2 presents the harmful score (HS) and finetuning accuracy (FA) across different cycle lengths and the corresponding number of samples used for updating the standard refusal feature. The results show that frequent updates with a short cycle help the Ref-Teacher model more effectively separate harmful from harmless prompts and generate appropriate refusal responses to harmful inputs.

## B.3 EFFECT OF REGULARIZATION STRENGTH ($\lambda$) ON REF-TEACHER MODEL TRAINING

The $\lambda$ value in Eq. 1 of main manuscript controls the strength of the regularization term that encourages distinct separation between the features of harmful and harmless prompts in the Ref-Teacher model during the teacher preparation stage. An overly strong regularization term may disrupt the internal representations of the Ref-Teacher model, while a weak regularization term may reduce the Ref-Teacher model's ability to distinguish between harmful and harmless prompts based on its refusal feature. Therefore, selecting an appropriate $\lambda$ value is critical for effective training of the Ref-Teacher model and subsequent finetuning. Table A3 presents the finetuning performance using Ref-Teacher models trained with different $\lambda$ values. The results show that a $\lambda$ value of 0.1 achieves the lowest harmful score (HS) and the highest finetuning accuracy (FA), indicating its effectiveness as an optimal hyperparameter choice.

## B.4 EFFECT OF THRESHOLD VALUES ON FINETUNING

The threshold $\tau$ in Eq. 2 is a key hyperparameter used as a standard to classify harmful prompts by measuring the similarity between input prompt features and the refusal feature in the Ref-Teacher model during the finetuning stage. We predicted prompts with similarity above the threshold as harmful, while those below the threshold are classified as harmless. Therefore, a threshold that is too low may misclassify harmful prompts as harmless, thereby introducing safety risks by allowing harmful prompts to be included in finetuning. Conversely, a threshold that is too high may incorrectly filter out harmless prompts misclassified as harmful, leading to reduced finetuning accuracy. As shown in Table A4, we evaluate the impact of varying threshold values. The results indicate that a threshold of 0.9 yields the lowest harmful score and the highest finetuning accuracy. This optimal performance is attributed to the near-perfect alignment of harmful prompt features with the refusal feature, resulting in the similarity values close to 1, in the Ref-Teacher model, as illustrated in Table 7 of the main manuscript.

## B.5 EFFECT OF ALIGNMENT DISTILLATION HYPERPARAMETERS

Knowledge distillation typically involves two key hyperparameters: temperature $T$, which controls the softness of the teacher predictions, and the distillation weight $\alpha$, which balances the influence of the distillation loss. To evaluate their impact, we measure both the harmful score and finetuning accuracy across various values of $T$ and $\alpha$. As shown in Table A5, higher values of $T$ lead to increased harmful scores, likely due to the student model not closely following the Ref-Teacher model's predictions. In contrast, higher values of $\alpha$ reduce the harmful score but also lower the fine-

Table A5: Impact of temperature ($T$) and distillation weight ($\alpha$) on Harmful Score (HS) and Fine-tuning Accuracy (FA). The best-performing setting ($T = 1.0$, $\alpha = 0.1$) is highlighted in bold.

| Temperature $T$ | $\alpha$ | HS ($\downarrow$) | FA ($\uparrow$) |
|---|---|---|---|
| **1.0** | **0.1** | **0.5** | **49.0** |
| 1.0 | 0.3 | 1.3 | 45.3 |
| 1.0 | 0.5 | 1.2 | 47.9 |
| 1.0 | 1.0 | 1.2 | 44.6 |
| 1.0 | 5.0 | 0.9 | 40.5 |
| 2.0 | 0.1 | 0.9 | 45.6 |
| 2.0 | 0.3 | 0.7 | 44.2 |
| 2.0 | 0.5 | 1.0 | 43.4 |
| 2.0 | 1.0 | 0.5 | 42.8 |
| 2.0 | 5.0 | 0.6 | 26.1 |
| 5.0 | 0.1 | 12.8 | 46.7 |
| 5.0 | 0.3 | 3.4 | 46.5 |
| 5.0 | 0.5 | 3.1 | 45.2 |
| 5.0 | 1.0 | 2.2 | 44.2 |
| 5.0 | 5.0 | 2.4 | 33.7 |

Table A6: Impact of data filtering on baseline. *HS* denotes Harmful Score (lower is better), and *FA* denotes Finetuning Accuracy (higher is better).

| Methods | No Filtering | | LLaMAGuard | | Ref-Teacher | |
|---|---|---|---|---|---|---|
| | HS | FA | HS | FA | HS | FA |
| SFT | 16.7 | 40.6 | 6.6 | 40.4 | 1.7 | 43.3 |
| RepNoise (Rosati et al. (2024)) | 30.4 | 38.4 | 13.2 | 37.2 | 2.5 | 36.7 |
| Vaccine (Huang et al. (2024d)) | 4.8 | 24.4 | 1.9 | 22.7 | 1.3 | 22.4 |
| Booster (Huang et al. (2024c)) | 5.9 | 43.4 | 3.2 | 43.7 | 0.9 | 44.2 |
| LDIFS (Mukhoti et al. (2023)) | 4.0 | 17.0 | 2.6 | 17.4 | 1.1 | 16.1 |
| Lisa (Huang et al. (2024b)) | 5.3 | 38.3 | 2.0 | 37.6 | 1.3 | 38.5 |
| Ref-Teacher (Ours) | **2.2** | **46.2** | **0.5** | **49.0** | **0.5** | **49.0** |

tuning accuracy, as excessive emphasis on the alignment loss weakens user-specific downstream task performance. Among these hyperparameter values, $T = 1$ and $\alpha = 0.1$ yield the best overall performance. This setting allows the student model to closely follow the well-aligned refusal responses of the Ref-Teacher model, while keeping the alignment loss moderate to preserve downstream task performance.

## C   ADDITIONAL EXPERIMENTS

### C.1   COMPARISON TO BASELINES WITH GUARDRAIL-BASED FILTERING.

Our proposed finetuning framework incorporates a data filtering process guided by the Ref-Teacher model, which is a fundamental defense against harmful finetuning attacks but has not yet been explored in the Finetuning-as-a-Service (FaaS) setting. To ensure that the superiority of our framework does not arise merely from data filtering, we additionally apply two filtering strategies, LLaMAGuard3-8B Llama Team (2024) and Ref-Teacher, to all baseline methods. Specifically, each baseline finetunes a safety-aligned model on user data filtered by (1) LLaMAGuard3-8B, which removes 5.7% of prompts, or (2) our Ref-Teacher filter, which removes 12.2% of promptswhen 100 harmful prompts are included among 1,000 user prompts. As shown in Table A6, both filtering methods reduces harmful scores across all baselines. Nevertheless, our framework consistently outperforms these improvements without relying on any external guardrail. This result is consistent with Table A10, where data filtering with Ref-Teacher achieves comparable safety gains but still falls short of the full effectiveness of our method.

### C.2   GENERALIZATION UNDER CROSS-DATASET FINETUNING

We conduct a cross-dataset evaluation to further assess generalization in the finetuning stage. Specifically, both the Ref-Teacher model and the safety-aligned models are trained on BeaverTails (Ji

Table A7: Cross-Dataset Evaluation (BeaverTails (Ji et al. (2023)) $\rightarrow$ JailbreakBench (Chao et al. (2024))). *HS* denotes Harmful Score (lower is better), and *FA* denotes Finetuning Accuracy (higher is better).

| Aligned Model | HS (In-Domain) ↓ | FA (In-Domain) ↑ | HS (Out-Domain) ↓ | FA (Out-Domain) ↑ |
|---|---|---|---|---|
| SFT | 16.7 | 40.6 | 93.0 | 40.6 |
| RepNoise (Rosati et al. (2024)) | 30.4 | 38.4 | 90.0 | 35.7 |
| Vaccine (Huang et al. (2024d)) | 4.8 | 24.4 | 15.0 | 23.6 |
| Booster (Huang et al. (2024c)) | 5.9 | 43.4 | 4.0 | 43.4 |
| LDIFS (Mukhoti et al. (2023)) | 4.0 | 17.0 | 81.0 | 17.0 |
| Lisa (Huang et al. (2024b)) | 5.3 | 38.3 | 9.0 | 35.7 |
| Ref-Teacher (Ours) | **0.5** | **49.0** | **2.0** | **46.6** |

Table A8: Performance on Llama3-8B-Instruct (Llama Team (2024)) under the pre-aligned LLM setting. Ref-Teacher* uses the raw instruct model as the Ref-Teacher without additional training, while Ref-Teacher denotes the model trained under our framework.

| Method | HS (↓) | FA (↑) |
|---|---|---|
| SFT | 64.0 | 66.0 |
| LDIFS (Mukhoti et al. (2023)) | 15.9 | **66.8** |
| SafeInstruct (Bianchi et al. (2023)) | 26.9 | 66.4 |
| Lisa (Huang et al. (2024b)) | 28.1 | 60.6 |
| Antidote (Huang et al. (2024a)) | 17.4 | 59.3 |
| Ref-Teacher* | 13.9 | 65.8 |
| Ref-Teacher | **5.4** | 66.5 |

et al. (2023)), and finetuning is then performed on JailbreakBench (Chao et al. (2024)). As shown in Table A7, several baselines suffer substantial performance degradation under this harmful data distribution shift, particularly in terms of harmfulness. In contrast, our Ref-Teacher-guided framework consistently achieves the lowest harmful scores and the highest finetuning accuracy in both in-domain and out-of-domain settings, demonstrating strong generalization across datasets.

## C.3 DISCUSSION OF THE PRE-ALIGNED LLM SETTING AND REF-TEACHER ADAPTATION

In practice, many safety-aligned LLMs are already available, such as Llama3-8B-Instruct Llama Team (2024), Gemma2-9B-it Team et al. (2024), and Qwen2-7B-Instruct Team (2024). However, following prior studies Huang et al. (2024b;d;a;c), we assume that such pre-aligned models are unavailable and begin from a base LLM. This assumption ensures a fair comparison with alignment-stage methods.

This setting is also realistic for new FaaS providers that have not yet established a safety-aligned model. These organizations must decide how to construct one that remains robust against harmful finetuning. They can either (1) adopt an alignment-stage approach or (2) perform standard supervised safety-alignment followed by a finetuning-stage defense. In this context, our framework introduces a specialized safety-aligned model that reliably identifies harmful prompts and provides alignment distillation.

In contrast, assuming a pre-aligned LLM simplifies our framework: the Ref-Teacher can be trained without updating the refusal feature or even be replaced by an existing safety-aligned model. Thus, while our main experiments target the more challenging scenario, our framework can be naturally extended to settings with pre-aligned models.

Moreover, pre-aligned LLMs are also not immune to safety degradation when directly finetuned on user data. Preventing this degradation again requires jointly finetuning on safety and user data, which also creates the gradient conflict. Therefore, a conflict-mitigation framework such as ours is needed even when starting from instruct models.

Therefore, to verify this, we evaluate all methods on Llama3-8B-Instruct. Since alignment-stage baselines cannot be applied, we compare only finetuning- and post-finetuning-stage methods. As shown in Table A8, our framework that adopting a pre-aligned LLM as Ref-Teacher achieves the lowest harmfulness score while preserving strong functional accuracy. Moreover, training the in-

Table A9: Performance comparison across different jailbreak attacks during finetuning. The GCG attack (Zou et al. (2023)) is generated using 100 samples from the BeaverTails dataset (Ji et al. (2023)), and the AutoDAN attack (Liu et al. (2023)) is generated using 520 samples from the AdvBench dataset (Zou et al. (2023)). The results demonstrate the strong safety alignment and generalization capability of our Ref-Teacher-guided finetuning strategy, which consistently outperforms all baselines.

| Methods | BeaverTails (Ji et al. (2023)) | | GCG (Zou et al. (2023)) | | AutoDAN (Liu et al. (2023)) | | Average | |
|---|---|---|---|---|---|---|---|---|
| | HS ↓ | FA ↑ | HS ↓ | FA ↑ | HS ↓ | FA ↑ | HS ↓ | FA ↑ |
| SFT | 16.7 | 40.6 | 36.0 | 40.6 | 69.6 | 40.6 | 40.8 | 40.6 |
| Repnoise (Rosati et al. (2024)) | 30.4 | 38.4 | 46.0 | 38.4 | 68.5 | 38.4 | 48.3 | 38.4 |
| Vaccine (Huang et al. (2024d)) | 4.8 | 24.4 | 16.0 | 24.4 | 18.3 | 24.4 | 10.4 | 24.4 |
| Booster (Huang et al. (2024c)) | 5.9 | 43.4 | 10.0 | 43.4 | 37.1 | 43.4 | 17.7 | 43.4 |
| LDIFS (Mukhoti et al. (2023)) | 4.0 | 17.0 | **4.0** | 17.0 | 61.9 | 17.0 | 23.3 | 17.0 |
| Lisa (Huang et al. (2024b)) | 5.3 | 38.3 | 52.0 | 38.3 | 41.5 | 38.3 | 32.9 | 38.3 |
| Ref-Teacher (Ours) | **0.5** | **49.0** | 6.0 | **49.0** | **0.9** | **49.0** | **2.5** | **49.0** |

Table A10: Effects of applying Ref-Teacher-guided finetuning to alignment-stage solutions.

| Methods | HS ↓ | FA ↑ |
|---|---|---|
| SFT | 16.7 | 40.6 |
| SFT+Ref-Teacher | **1.1** | **42.1** |
| Repnoise (Rosati et al. (2024)) | 30.4 | 38.4 |
| Repnoise+Ref-Teacher | **1.4** | **39.2** |
| Vaccine (Huang et al. (2024d)) | 4.8 | **24.4** |
| Vaccine+Ref-Teacher | **2.2** | 22.0 |
| Booster (Huang et al. (2024c)) | 5.9 | 43.4 |
| Booster+Ref-Teacher | **1.9** | **43.8** |

struct model as a Ref-Teacher further enhances safety. These results confirm that our approach remains effective not only for base models but also for pre-aligned models, offering additional safety benefits.

## C.4 ROBUSTNESS AGAINST ADVANCED JAILBREAKING ATTACK

When jailbreaking LLMs, advanced techniques such as GCG (Greedy Coordinate Gradient)[1] (Zou et al. (2023)) and AutoDAN (Automatically generating DAN-series-like jailbreak prompts)[2] (Liu et al. (2023)) can be used to induce harmful responses beyond simply prompting with harmful queries. These methods demonstrated a high attack success rate in eliciting harmful responses, even from safety-aligned models, compared to direct harmful prompts. To evaluate the robustness of our Ref-Teacher-guided finetuning strategy against such advanced jailbreaking attacks, we measure harmful score under both GCG and AutoDAN attacks, targeting Llama3-8B-Instruct in a black-box setting. While all methods show increased harmful scores under these advanced attacks, Table A9 demonstrates that our Ref-Teacher-guided finetuning method is more robust than baseline approaches. Notably, although the LDIFS method achieves a low harmful score under the GCG attack, it suffers from poor finetuning accuracy and exhibits a high harmful score under the AutoDAN attack, supporting its impracticality. In contrast, our method maintains both a low harmful score and high finetuning accuracy under both GCG and AutoDAN attacks, demonstrating its effectiveness in providing reliable protection against increasingly sophisticated jailbreak attempts.

## C.5 REINFORCING ALIGNMENT-STAGE SOLUTIONS WITH REF-TEACHER-GUIDED FINETUNING STRATEGY.

To identify whether our Ref-Teacher-guided finetuning strategy can further enhance the safety and user-specific task performance of safety-aligned models from alignment-stage techniques, we apply our method to these aligned models during finetuning stage and measure both the harmful score

---

[1]https://github.com/GraySwanAI/nanoGCG

[2]https://github.com/SheltonLiu-N/AutoDAN

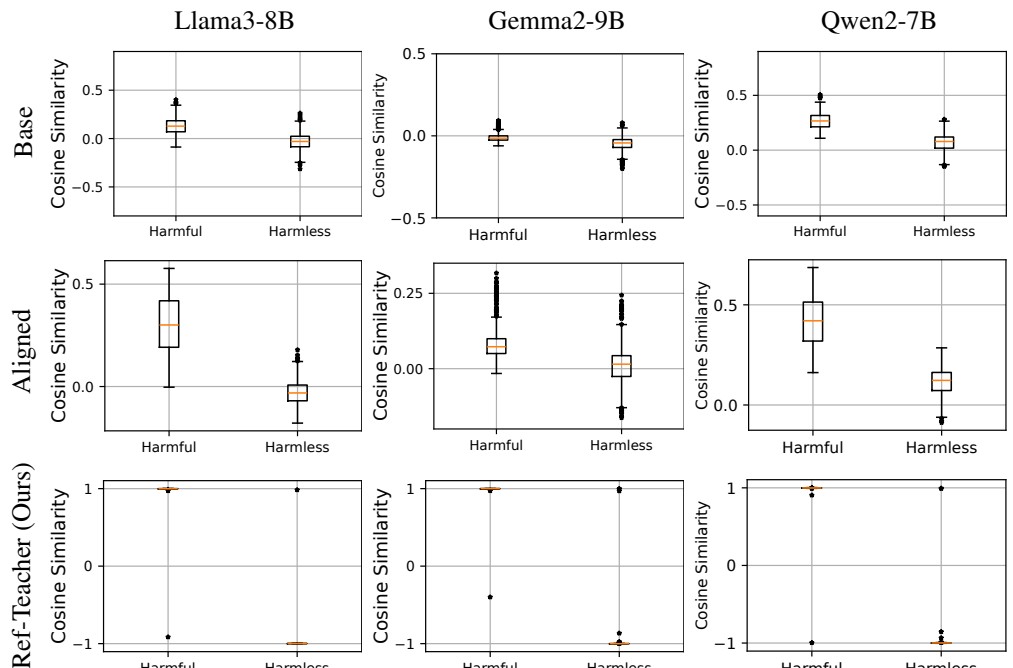

Figure A3: Box plot of cosine similarity distributions for harmful and harmless prompts in the base model, aligned model, and Ref-Teacher (Ours). Prompts were sampled from the BeaverTails (harmful, n=500) and Alpaca (harmless, n=500) datasets, representing diverse general prompts. The sampled prompts visualized here were excluded from the Ref-Teacher training set. This visualization highlights that safety-alignment introduces the capability to distinguish harmful from harmless prompts.

Table A11: Accuracy of classifying prompts using refusal features. Prompts with cosine similarity above the threshold are classified as harmful, while those below are classified as harmless.

| Model | Threshold | Harmful Acc | Harmless Acc | Total Acc |
|---|---|---|---|---|
| Llama3-8B | 0.34 | 86.0% | 78.8% | 82.4% |
| Llama3-8B-Instruct | 0.06 | 95.2% | 93.6% | 94.4% |
| Llama3-8B-Ref-Teacher | 0.97 | 99.8% | 99.8% | 99.8% |
| Gemma2-9B | -0.037 | 87.8% | 61.2% | 74.5% |
| Gemma2-9B-Instruct | 0.035 | 90.4% | 70.4% | 80.4% |
| Gemma2-9B-Ref-Teacher | 0.97 | 99.8% | 99.6% | 99.7% |
| Qwen2-7B | 0.15 | 97.6% | 88.8% | 93.2% |
| Qwen2-7B-Instruct | 0.24 | 93.2% | 97.2% | 95.2% |
| Qwen2-7B-Ref-Teacher | 0.9 | 99.8% | 99.6% | 99.7% |

(HS) and finetuning accuracy (FA). As shown in Table A10, our approach significantly reduces the harmful score while maintaining comparable finetuning accuracy in most cases. The reinforced safety-alignment demonstrates that Ref-Teacher-based data filtering and alignment distillation can complement the alignment-stage solutions. However, the performance of this setting remains inferior to our finetuning framework, highlighting the importance of directly finetuning the base model under Ref-Teacher guidance.

# D  SAFETY ALIGNMENT ENDOWS MODELS WITH REFUSAL-BASED HARMFULNESS DETECTION

Safety-aligned LLMs tend to exhibit distinct response behaviors as input prompts vary in harmfulness, and this tendency is reflected in their refusal feature, which can serve as a signal for harmfulness classification. While base models can sometimes provide a weak discriminative signal, we observe that this property is more pronounced and reliable in safety aligned models.

To validate this hypothesis, we measure the cosine similarity between the feature of each input prompt and a refusal feature in both base and safety-aligned models, and then assess whether harm-

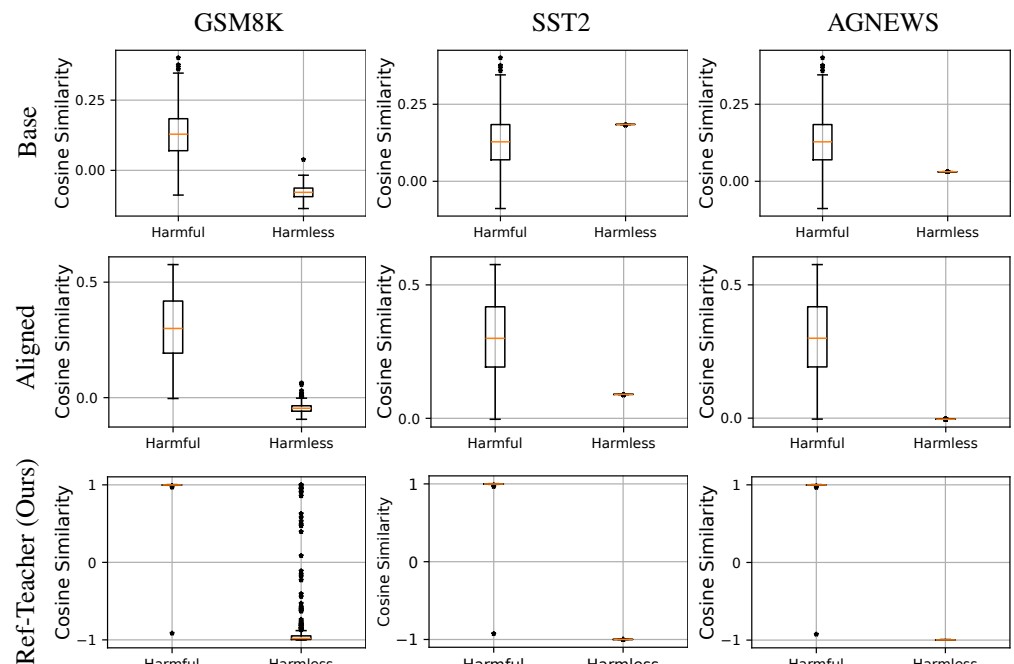

Figure A4: Box plot of cosine similarity distributions for harmful and harmless prompts, evaluated on the base model, aligned model, and Ref-Teacher (Ours). Harmful prompts were sampled from the BeaverTails dataset ($n = 500$), while harmless prompts were sampled from GSM8K, SST2, and AGNEWS ($n = 500$), which are domain-specific downstream task datasets used during the finetuning stage.

Table A12: Classification accuracy using refusal features. Prompts with cosine similarity above the threshold are identified as harmful, and those below as harmless. Thresholds are optimized to maximize total classification accuracy.

| Datasets | Model | Threshold | Harmful Acc | Harmless Acc | Total Acc |
|---|---|---|---|---|---|
| GSM8K | Llama3-8B | -0.017 | 95.6% | 99.8% | 97.7% |
| | Llama3-8B-Instruct | 0.035 | 98.2% | 99.6% | 98.9% |
| | Llama3-8B-Ref-Teacher | 0.965 | 99.8% | 99.2% | 99.5% |
| SST2 | Llama3-8B | 0.190 | 22.6% | 100.0% | 61.3% |
| | Llama3-8B-Instruct | 0.095 | 89.6% | 100.0% | 94.8% |
| | Llama3-8B-Ref-Teacher | -0.920 | 100.0% | 100.0% | 100.0% |
| AGNEWS | Llama3-8B | 0.032 | 86.0% | 100.0% | 93.0% |
| | Llama3-8B-Instruct | 0.010 | 99.8% | 100.0% | 99.9% |
| | Llama3-8B-Ref-Teacher | -0.990 | 100.0% | 100.0% | 100.0% |

ful and harmless prompts can be separated on the refusal feature. Figure A3 shows the resulting distributions for BeaverTails (harmful) (Ji et al. (2023)) and Alpaca (harmless) (Taori et al. (2023)). Safety-aligned models yield more clearly separated similarity distributions, enabling more reliable discrimination, whereas base models exhibit substantial overlap, though not complete indistinguishability. Numerical results in Table A11 confirm this trend, safety-aligned models achieve higher classification accuracy than the base models for both harmful and harmless prompts.

We further extend the analysis to GSM8K (Cobbe et al. (2021)), SST2 (Socher et al. (2013)), and AGNEWS (Zhang et al. (2015)), which are used during finetuning. Following the same setup as in Fig. A3 and Table A11, we use BeaverTails as harmful data and GSM8K, SST2, and AGNEWS as harmless data with LLaMA3-8B (Llama Team (2024)). Figure A4 reports cosine similarity distributions and Table A12 reports accuracy using the optimal threshold per dataset. Since these downstream datasets are domain-specific and differ from BeaverTails in distribution, the base model shows some separability. Nevertheless, safety-aligned models consistently produce clearer separation and higher accuracy, and Ref-Teacher yields the most distinct separation and the strongest classification performance.

## E    LIMITATION

Our Ref-Teacher-guided finetuning framework relies on the Ref-Teacher model, which is trained using the refusal feature. Consequently, its safety-alignment could be compromised if adversarial attacks are designed to disrupt or manipulate the refusal feature. In such cases, the customized model finetuned under the guidance of a compromised Ref-Teacher may also inherit weakened safety-alignment.

## F    LLM USAGE

Large Language Models (ChatGPT-5) were used only for improving grammar and clarity in writing. They did not contribute to research ideation, experimental design, or analysis.

