# OpenReview forum: "Safety-Aligned Weights Are Not Enough: Refusal-Teacher-Guided Finetuning Enhances Safety and Downstream Performance under Harmful Finetuning Attacks"
_ICLR.cc/2026/Conference — ICLR 2026 Conference Withdrawn Submission_

### Official Review · Reviewer_MKsJ · 2025-11-01

**Soundness:** 3
**Presentation:** 3
**Contribution:** 3
**Rating:** 6
**Confidence:** 4

**Summary:**

This paper addresses maintaining safety alignment in LLMs under Finetuning-as-a-Service (FaaS) against malicious finetuning attacks. It finds that the standard “safety-align then finetune” pipeline weakens performance, while joint finetuning causes gradient conflicts between safety and task goals. To fix this, the authors propose Refusal-Teacher (Ref-Teacher) finetuning. Results show improved safety and task accuracy across multiple settings.

**Strengths:**

1. The paper's motivation is clearly shown through the experiments in Section 4, which effectively frame the problem the proposed method aims to solve.
2. The experimental evaluation is comprehensive, covering a diverse range of datasets and settings.
3. The paper is well-written and easy to follow.

**Weaknesses:**

1. My main concern is the fairness of the comparison. The proposed method uses a data filtering step that the baselines lack, and this filter appears optimized for the evaluation tasks. Although Appendix C1 includes a related comparison with LLaMAGuard3-8B, a more direct evaluation applying the same trained data filter to the baseline methods is needed.
2. The method adds several components that likely increase computational cost and deployment complexity. The paper would benefit from a clear analysis of the time and memory overhead relative to the baselines.

**Questions:**

1. Could you provide a detailed analysis of the computational cost (e.g., training time, memory usage) of your method compared to the standard SFT and other finetuning-stage baselines?
2. The "Base -> SA + FT" method mentioned in Section 4 appears to be a strong baseline. What was the reasoning for not including it in the main comparison tables?

---

> ### Author Response · Authors · 2025-11-21
>
> We thank the reviewer for the insightful feedback. The points on fairness and computational analysis are particularly valuable and will guide our revisions. All tables and references are placed at the top of this page due to space constraints.
> ## **Fairness of Comparison (W1)**
> Our key contribution is a novel FaaS framework that jointly trains the base LLM on safety and user data while mitigating gradient conflicts. Harmful samples exacerbate these conflicts, which is why **data filtering is only one component of our framework, not the main contribution.** As shown in Table 9, filtering improves safety, but even without filtering, **alignment distillation alone already outperforms prior state-of-the-art methods.**
>
> We also provide comparisons where filtering is added to existing baselines (Tables A6 and A10). Although filtering helps these baselines, **they still do not outperform our method.** Notably, Table 3 shows that even when a baseline is trained with no harmful data (p = 0), it still performs worse than our method at p = 0.1. This highlights that **our gains are not merely due to filtering, but rather to the structural design of our framework.**
>
> Nevertheless, to directly address the reviewer’s concern, we additionally apply **LlamaGuard3-8B finetuned on BeaverTails** as a trained filtering model to all baselines. Table R7 reports its **classification accuracy**, and Table R8 shows the resulting **harmfulness and finetuning accuracy.**
>
> The results show the following:
> 1. Table R7: LlamaGuard is well-finetuned and achieves high filtering accuracy.
> 2. Table R8: Even when all baselines use the in-domain filtering mechanism, Ref-Teacher still achieves the lowest harmful scores and highest finetuning accuracy
>
> This confirms that the strength of our approach does not come from filtering alone, but from **the combined design of joint finetuning and alignment distillation under gradient conflict mitigation.**
>
> We will **include all filtered-baseline comparisons in the main tables** and provide the LlamaGuard finetuning details in the camera-ready version.
> ## **Computational Cost (W2, Q1)**
> To quantify the computational overhead of our framework, we measured both GPUTime and GPUMemory for the alignment and finetuning stage separately using four RTX 3090 GPUs. Table R1 reports the per-GPU GPUTime (average per-step runtime) and GPUMemory (average per-step memory usage).
>
> As shown in Table R1, our method incurs additional cost relative to SFT, but **the increase remains moderate compared to other baselines.** Specifically, compared to SFT, our framework requires 44.0% more GPUTime and 35.9% more GPUMemory, yet it achieves a 93.8% reduction in harmful score and a 22.8% increase in finetuning accuracy.
>
> Moreover, when compared to Booster, state-of-the-art baseline, Ref-Teacher uses 34.7% less GPUTime and 24.6% more GPUMemory, while achieving an 83.1% reduction in harmful score and an 11.4% improvement in finetuning accuracy.
>
> Together, these results show that **the computational overhead introduced by Ref-Teacher is modest and well-justified**, given the substantial safety and utility improvements provided by our framework.
> ## **Baseline Inclusion (Q2)**
> The "Base → SA + FT" setting corresponds to the core training pipeline used in our method, where we apply safety alignment (SA) and downstream finetuning (FT) jointly on the base model. Because this procedure is an internal component of our proposed framework rather than an independent competing baseline, we did not list it separately in the main tables.
>
> However, we agree that including this variant would help clarify the contribution of each component in our framework. We will therefore add it as an additional baseline in the main comparison tables in the camera-ready version.

---

### Official Review · Reviewer_B6XU · 2025-11-01

**Soundness:** 2
**Presentation:** 3
**Contribution:** 3
**Rating:** 4
**Confidence:** 5

**Summary:**

This paper addresses the problem of maintaining LLM safety during downstream finetuning. The authors identify a key limitation in the standard two-stage pipeline where a safety-aligned model is first prepared and then finetuned on user data. They then argue that this approach leads to suboptimal downstream task performance due to ‘weak initialization’ and can still compromise safety. As an alternative, the authors propose a Refusal-Teacher (Ref-Teacher)-guided finetuning framework. Instead of finetuning a pre-aligned model, this framework directly finetunes the base LLM. The process is guided by a specially trained, frozen Ref-Teacher model. This teacher serves two functions: 1) it filters harmful prompts from the user's data using a learned *refusal feature*, and 2) it provides *alignment distillation* by generating soft refusal labels for a separate safety dataset, which helps the student model learn safety objectives with reduced gradient conflicts. The authors demonstrate that their framework consistently outperforms existing methods, achieving lower harmfulness scores while simultaneously attaining higher accuracy on downstream tasks.

**Strengths:**

- The paper is well-written and easy to follow. The motivation is laid out logically, building a clear case for why a new approach is needed.
- The proposed solution, to finetune the base model directly while carefully managing the safety/utility trade-off, is an effective response to this finding. This work provides a practical approach to a problem in Finetuning-as-a-service.

**Weaknesses:**

- The data filtering strategy is configured to maximize recall on harmful prompts, which may discard some harmless user data (a high false positive rate). It is unclear what the percentage of harmless data filtered out is across different tasks.
- The authors should experiment with other data filtering methods [1][2] for a more comprehensive comparison.
- [3][4] also studies this problem from the similarity perspective, which should also be discussed in the revision.

[1] Deep ignorance: Filtering pretraining data builds tamper-resistant safeguards into open-weight LLMs

[2] Pharmacist: Safety Alignment Data Curation for Large Language Models against Harmful Fine-tuning

[3] Why LLM Safety Guardrails Collapse After Fine-tuning: A Similarity Analysis Between Alignment and Fine-tuning Datasets

[4] When Style Breaks Safety: Defending Language Models Against Superficial Style Alignment

**Questions:**

- Could the authors provide a brief analysis of the computational overhead of the ‘Teacher Preparation Stage’ compared to the standard ‘Alignment Stage’ used for the baselines.

- Could an adversary craft finetuning examples that are benign in their feature representation (low cosine similarity to the refusal feature) but still steer the model towards unsafe behavior on related prompts?

---

> ### Author Response · Authors · 2025-11-21
>
> We appreciate the reviewer’s careful evaluation and valuable feedback. Due to space constraints, all tables and references appear at the top of this page.
> ## **Percentage of Harmless Data Filtered Out (W1).**
> We report the classification accuracy of harmful and harmless prompts during finetuning in Table 7, which shows that **the Ref-Teacher achieves near-perfect discrimination** on GSM8K, SST2, and AGNEWS. This implies that **few harmless user data is mistakenly filtered out.**
>
> For AlpacaEval, about 23% of the data is filtered out, yet Table 5 confirms that performance remains strong. This is because **our improvements do not rely on filtering alone but largely stem from the structural improvement of the FaaS paradigm.**
> ## **Comparison with Previous Data Filtering Methods (W2)**
> While our experiments primarily focused on alignment- and finetuning-stage solutions, and we agree that comparing against **Deep Ignorance** and **Pharmacist** strengthens the evaluation.
> - **Deep Ignorance** originally filters pretraining data using a blocklist and a ModernBERT-based classifier. However, **for a fair comparison with our work, we instead apply its data filtering to user data during finetuning.**
> - **Pharmacist** selects high-quality safety-critical data for safety-alignment. **Since it is not designed to distinguish harmful vs. harmless prompts, we evaluate it following its original setup, alignment-stage solution.**
>
> Table R6 reports the harmfulness classification accuracy of Deep Ignorance, and Table R7 compares our method with baselines including Deep Ignorance and Pharmacist.
> - Deep Ignorance: Despite being trained on the same data as Ref-Teacher, it shows low harmful prompt detection accuracy (Table R6), which leads to higher harmful scores in Table R7.
> - Pharmacist: Table R7 demonstrates that alignment-stage filtering alone cannot defend against harmful user data during finetuning.
> - Ours: It performs **accurate data filtering during the finetuning stage**, allowing us to achieve better safety and utility.
>
> **We will include Deep Ignorance and Pharmacist as additional baselines** in the camera-ready version.
> ## **Additional Related Works (W3)**
> Prior work [2] analyzes how safety guardrails collapse when the representation of downstream data is similar to the safety-alignment data, and another work [3] shows that applying benign prompt styles to harmful prompts can break safety, proposing a style-augmentation method to mitigate this issue. Both studies are conceptually related to our refusal-feature-similarity-based data filtering, as they also investigate how feature-space-similarity between harmful and benign distributions contributes to safety degradation. Therefore, **we will include a discussion of these works in the revised Related Work section.**
> ## **Computational Overhead (Q1)**
> To quantify the computational overhead introduced by the Teacher Preparation Stage, we measured both **GPUTime and GPUMemory** for the alignment stage and the finetuning stage separately. All measurements were performed on four RTX 3090 GPUs, and Table R1 reports the per-GPU GPUTime (average per-step runtime) and GPUMemory (average per-step memory usage).
>
> As shown in Table R1, in the teacher preparation (alignment) stage, our method requires 0.85s more GPUTime and 3.45GB more GPUMemory compared to SFT. While Ref-Teacher does incur additional cost relative to SFT, the increase is moderate compared to other baselines. Specifically, compared to SFT, Ref-Teacher uses 44.0% more GPUTime and 35.9% more GPUMemory, yet achieves a 93.8% reduction in harmful score and a 22.8% increase in finetuning accuracy. These results indicate that **the computational overhead is modest and well-justified given the substantial safety and utility improvements delivered by our framework.**
> ## **Adversarial Examples with Low Cosine Similarity to the Refusal Feature (Q2)**
> As discussed in the Limitation (Appendix E), our data filtering method relies on the cosine similarity between the refusal feature and the prompt feature. Therefore, it is theoretically possible for crafting adversarial examples that exhibit low cosine similarity to the refusal feature while still encoding adversarial intent, in which case the filtering step may fail. However, **creating such prompts while also keeping them user-intended and natural enough to pass perplexity checks is expected to be highly challenging in practice.**
>
> More importantly, **our framework does not depend solely on data filtering**. As shown in Table 7, data filtering is not perfect during finetuning, yet the overall performance remains strong. Furthermore, Table 9 shows that even without filtering, the model remains robust because the framework jointly learns from safety data while mitigating gradient conflict during finetuning. This demonstrates that **Ref-Teacher is resilient to imperfect filtering and maintains safety even when some adversarial examples bypass the similarity-based filter.**

---

> > ### Comment · Reviewer_B6XU · 2025-11-25
> >
> > Thank you for the detailed rebuttal. It mostly addresses my earlier concerns. Especially the inclusion of Deep Ignorance and Pharmacist as additional baselines further strengthens the evaluation. I also appreciate the clarification that the framework’s robustness does not rely on perfect filtering, which further demonstrates the robustness of the method.
> >
> > I look forward to seeing the updated experiments and expanded discussions in the final camera-ready version. I have raised my score.

---

> > > ### Author Response · Authors · 2025-11-25
> > >
> > > Dear Reviewer B6XU,
> > >
> > > We appreciate your follow-up and are glad to hear that our responses have adequately addressed your concerns. Thank you for taking the time to review our work and share your feedback, which helped us clarify and strengthen our evaluation.
> > >
> > > Best regards, Authors of Paper 8215

---

### Official Review · Reviewer_CsFX · 2025-11-01

**Soundness:** 2
**Presentation:** 2
**Contribution:** 1
**Rating:** 2
**Confidence:** 4

**Summary:**

The paper focuses on the problem of fine-tuning degrading safety alignment in language models. The setting is fine-tuning-as-a-service, where a user wants to fine-tune a model on some task-specific data. The user is assumed adversarial such that a certain fraction of user data consists of harmful prompts and harmful responses.

The paper proposes to first train, starting from a base model, a safety-aligned teacher model called Ref-Teacher. Then, in the fine-tuning stage, Ref-Teacher is used in two ways: 'refusal feature' from the teacher is used for classifying whether user prompts are harmful (and filtering out use samples classified as harmful); and using soft 'refusal labels' from the teacher for balancing loss on user data and KL-divergence loss on alignment data (called as alignment distillation).

Experiments consider fine-tuning 3 base models on 4 tasks, and show that the proposed method outperforms several baselines on the metrics of fine-tune accuracy and harmful score.

**Strengths:**

* The problem of fine-tuning degrading safety alignment is practically relevant and has recently received a widespread attention.
* The idea of using signals from a safety-aligned teacher model is interesting.

**Weaknesses:**

* Quite a large number of solutions have been recently proposed for mitigating safety degradation after fine-tuning. Besides alignment stage defenses (baselines in the paper, such as Vaccine and Booster, fall under this), there are fine-tuning-stage defenses (e.g., SafeInstruct [Bianchi et al., 2024], VLGuard [Zong et al., 2024], constrained-SFT [Qi et al., 2024]), and post-fine-tuning defenses (e.g., SafeLoRA [Hsu et al., 2025], RESTA [Bharadwaj et al., 2024], SOMF [Yi et al., 2024],  Antidote [Huang et al., 2024]). The paper does not acknowledge the vast related work on this topic. While it is infeasible to compare against too many baselines, it is important to acknowledge the related work on this topic and provide qualitative comparisons. For fairness of comparison, it will be great if the authors can consider a couple of baselines from other setups (e.g., one from post-fine-tuning-stage and one from fine-tuning-stage) for comparison.

* One of my main concerns is that the paper takes an overly simplistic approach for baselines for preserving alignment after fine-tuning -- first alignment is performed by supervised fine-tuning (SFT) of a base model on alignment data and then task-specific fine-tuning is performed. In practice, users significantly prefer fine-tuning instruct models. Leading instruct models take a number of steps for alignment beyond simple SFT on alignment data including RLHF via preference tuning such as Direct Preference Optimization (DPO) or other online RL algorithms such as Proximal Policy Optimization (PPO). Consequently, when starting from an instruct model, adaptation to the user task is often easier and the safety degradation is typically less significant. Many prior works on the topic of fine-tuning degrading safety alignment consider the case of adapting instruct (or chat) models and the impact of safety alignment (e.g., constrained-SFT [Qi et al., 2024], SafeLoRA [Hsu et al., 2025]). The paper has limited experiments in Appendix C.3 when considering instruct models as Ref-Teacher, but lacks details on using instruct models as starting points for task specific fine-tuning.

* The paper does not conduct any ablation experiments to quantify the contributions of data filtering and alignment distillation. (More details in the Questions.)

References
1. T. Huang, G. Bhattacharya, P. Joshi, J. Kimball, L. Liu, "Antidote: Post-fine-tuning Safety Alignment for Large Language Models against Harmful Fine-Tuning", 2024
2. Tiansheng Huang, Sihao Hu, Fatih Ilhan, Selim Furkan Tekin, Ling Liu,"Booster: Tackling Harmful Fine-tuning for Large Language Models via Attenuating Harmful Perturbation", 2024
3. Federico Bianchi, Mirac Suzgun, Giuseppe Attanasio, Paul Röttger, Dan Jurafsky, Tatsunori Hashimoto, and James Zou, "Safety-Tuned LLaMAs: Lessons From Improving the Safety of Large Language Models that Follow Instructions", 2024
4. Yongshuo Zong, Ondrej Bohdal, Tingyang Yu, Yongxin Yang, and Timothy Hospedales, "Safety fine-tuning at (almost) no cost: A baseline for vision large language models", 2024
5. Xiangyu Qi, Ashwinee Panda, Kaifeng Lyu, Xiao Ma, Subhrajit Roy, Ahmad Beirami, Prateek Mittal, and Peter Henderson, "Safety alignment should be made more than just a few tokens deep", 2024
6. Chia-Yi Hsu, Yu-Lin Tsai, Chih-Hsun Lin, Pin-Yu Chen, Chia-Mu Yu, and Chun-Ying Huang, "Safe LoRA: the Silver Lining of Reducing Safety Risks when Fine-tuning Large Language Models", 2025
7. Rishabh Bhardwaj, Do Duc Anh, and Soujanya Poria, "Language models are homer simpson! safety re-alignment of fine-tuned language models through task arithmetic", 2024
8. Xin Yi, Shunfan Zheng, Linlin Wang, Xiaoling Wang, and Liang He, "A safety realignment frame- work via subspace-oriented model fusion for large language models", 2024

**Questions:**

* The proposed method has two stages so-called alignment distillation and data filtering. It is not clear how much each stage contributes and how two stages help each other. Are there any ablation experiments quantifying the contribution of each stage? For instance, data filtering can be used in the conventional setup of filtering on top of an aligned model. How would it compare with the proposed method?
* In the experiment, FA is measured on downstream benchmarks by using specific number of samples (L359 on page 7). Why specific number of samples are chosen from these benchmarks in contrast to using the entire test set?
* In Algorithm 1, L260, the equation number (eq (2)) seems to be a typo.
* In Table 8, including BeaverTails seems a bit unfair since Ref-Teacher is trained on BeaverTails. Can the author give more details on the inclusion of BeaverTails?

---

> ### Author Response · Authors · 2025-11-21
>
> We are grateful for the reviewer’s constructive feedback. All tables and references appear at the top due to space limitations.
> ## **Additional Prior Works (W1)**
> While our paper **already compares against alignment-stage (RepNoise, Vaccine, Booster) and finetuning-stage (LDIFS, Lisa) approaches**, we agree that more baselines strengthens the evaluation. Therefore, we **additionally evaluate SafeInstruct (finetuning–stage solution) and Antidote (post-finetuning solution)**, as shown in Table R2.
>
> Table R2 indicates that **our Ref-Teacher–guided framework outperforms these additional baselines** across alignment-, finetuning-, and post-finetuning-stage defenses.
> We will include SafeInstruct and Antidote in the camera-ready version.
> ## **Instruct LLMs (W2)**
> We **follow the same assumption used in prior works** such as Booster, Vaccine, Lisa, and Antidote, which also start from a base LLM rather than assuming access to a safety-aligned model. This ensures a fair comparison with alignment-stage methods.
>
> This assumption is practical in scenarios where **new FaaS providers do not yet prepare a safety-aligned model**. Such organizations must decide how to construct one that is robust against harmful finetuning. They can choose (1) an alignment-stage solution, or (2) standard supervised safety-alignment with a finetuning-stage solution. In this setting, we propose a specialized safety-aligned model that accurately distinguishes harmful prompts.
>
> Importantly, **instruct models also suffer safety degradation when directly finetuned on user data.** Preventing this degradation again requires finetuning safety and user data together, which also create the gradient conflict. Therefore, a conflict-mitigation framework such as ours is needed even when starting from instruct models.
>
> Moreover, **assuming that a pre-aligned model already exists is actually an easier setting.** If available, Ref-Teacher can be trained without updating the refusal feature or even replaced by the existing safety-aligned model (Table A8). Thus, while we target the more challenging scenario, **our framework naturally extends to cases where a safety-aligned model is already present.**
>
> Nevertheless, to address the reviewer’s concerns, **we evaluate all methods on Llama3-8B-Instruct**. Since alignment-stage baselines cannot be applied, we compare only finetuning- and post-finetuning-stage solutions. As reported in Table R3, our approach achieves substantially lower harmful scores while maintaining competitive accuracy, and training the instruct model as Ref-Teacher further improves safety.
> These results show that our framework is effective not only for base models but also for instruct models, providing additional safety benefits even when starting from strong initial checkpoints.
> ## **Ablation Study (W3, Q1)**
> Since **the individual contributions of alignment distillation and data filtering are already quantified in Tables 9 and 10**, we interpret the reviewer’s concern as asking how baselines perform when equipped with our Ref-Teacher–guided filtering. Although Table A10 reports results for three baselines, we extend this evaluation to all baselines in Table R4.
>
> Table R4 shows that **our filtering improves all baselines, but none surpasses our full framework.** This indicates that the gains of our method do not stem from filtering alone. Rather, they arise from the combination of alignment distillation and conflict-mitigated joint training, which represents a **structural improvement to the FaaS paradigm**.
> ## **Test sets (Q2)**
> We **follow the evaluation protocol used in prior works**, including Booster, Vaccine, Lisa, and Antidote, all of which evaluate accuracy on sampled subsets. To ensure fair comparison, we adopt the same sample sizes.
>
> **For AlpacaEval and SST2, our evaluation already uses the full test sets.** But there was a typo in reporting the test set sizes of SST2 and GSM8K; SST2 is evaluated on its validation set of 872 samples. We will correct this in the revised version.
>
> To directly address the reviewer’s concern, we evaluate all methods on the full test sets of GSM8K and AGNEWS in Table R5.
> As a result, **the relative performance trends remain unchanged, confirming that the sample-based protocol does not affect our conclusions.**
> ## **Typo (Q3)**
> Thank you for catching this. We will correct this in the revised version.
> ## **BeaverTails in Table 8 (Q4)**
> As noted in Section 6.2, **Ref-Teacher is trained on Alpaca as harmless data, while Table 8 evaluates models using the harmless portion of JailbreakBench.** Thus, the "BeaverTails" results in Table 8 correspond to a setting where only the harmless portion is replaced. Although harmful data in "BeaverTails" is an in-domain dataset, it is not part of our training set. Consequently, Ref-Teacher shows consistent superiority across multiple harmful datasets, indicating that its performance is not driven by overlap with training data.

---

### Official Review · Reviewer_jpSC · 2025-11-03

**Soundness:** 2
**Presentation:** 2
**Contribution:** 2
**Rating:** 4
**Confidence:** 3

**Summary:**

The paper proposes Ref-Teacher fine-tuning method, where a base model is fine-tuned on downstream tasks along with safety data to preserve its safety alignment properties. The paper first makes the observation that directly fine-tuning the base model achieves both robust safety-alignment and good downstream task performance, but can also introduce "gradient conflicts" where the safety and task gradient directions may be at odds with each other, resulting in worse task performance compared to training on the task only. They then train a new teacher model called Ref-Teacher by filtering out harmful prompts via refusal alignment and training on a newly proposed regularized loss function. The Ref-Teacher is then used to train a distilled model.

**Strengths:**

1. The paper tackles a relevant, current and pervasive problem.
2. The experimental section shows clear improvements over state-of-the-art

**Weaknesses:**

1. The solution seems costly, and it is not clear if it is worth the benefits for LLMs where training is already prohibitively expensive.
2. The solution is non-intuitive and relatively more complex in terms of implementation vs other state-of-the-art ones
3. It requires changing the data itself, raising concerns about distributional shifts and making the generalizability questionable.
4. The depth of the novelty is not clear, and quite a few of the observations such as training on both safety and task datasets, gradient differences between them, etc. are already well known in practice.

**Questions:**

1. Why did the authors take the route of creating a Teacher model which both creates harmful/harmless prompts and is distilled from? Is it simply because empirically it gives better results or is there an intuition behind this? The state-of-the-art papers (such as RepNoise and Lisa) have very elegant solutions compared to the 3 to 5 step process that the authors have employed here.
2. While the results from the experimental section show that they are better, the main question that the paper does not properly answer is why. Why must the Ref-Teacher be used for more effectively distinguishing harmful vs harmless prompts? Why not use a better model?
3. What are the advantages of this compared to using the standard methods of detecting harmful prompts?
4. "we assume a setting where a pre-aligned model is unavailable" - this is a strong assumption. Can you please explain a practical scenario where this might be the case?
5. This solution seems to be a two-step solution - creating a teacher model to distill from, and also creating new filtered safety-targeted dataset to train on. The state-of-the-art frameworks compared against do not do training on a new safety-targeted dataset but use the base datasets. Is that not a concern since the solution seems dependent on the underlying data too to a certain extent?
6. What are the resource costs of this solution in terms of number of extra FLOPs? Is the cost worth the performance gains and/or implementation complexities?

---

> ### Author Response · Authors · 2025-11-21
>
> We appreciate the reviewer’s thorough review and helpful suggestions. All tables and references are included at the top of this page due to space limits.
> ## **Motivations & Novelty (W2, W4, Q1)**
> While some observations are already known, our contribution does not lie in rediscovering them. Instead, we **identify a structural flaw in the conventional FaaS paradigm** and propose a **novel unified framework** that resolves this issue.
>
> Traditionally, most of existing FaaS methods follow a two-stage process: (1) safety-align the base model, and (2) finetune this safety-aligned model on user data.
>
> However, this practice is **sub-optimal**. A safety-aligned model's parameters are biased toward safety behaviors, which **hinder effective adaptation to downstream tasks**.
>
> In contrast, our framework **jointly optimizes the base LLM** on both user data and safety-alignment data, providing a **more adaptable initialization**. This strategy yields stronger safety and utility (Table 1), but it introduces gradient conflict that worsen under harmful user data (Table 2). To address this, we designed a single unified model, the **Ref-Teacher**, which performs both alignment distillation and data filtering to mitigate gradient conflicts.
>
> Importantly, **our approach does not introduce extra stages or excessive complexity**; it simply replaces the alignment stage with a teacher-preparation stage, resulting in a conceptually cleaner.
> ## **Ref-Teacher-guided Data Filtering (Q2, Q3)**
> Our filtering approach is designed to fully leverage the properties of the **Refusal Feature**, which shows **high cosine similarity with harmful prompt features** and **low similarity with harmless ones**.
> We train the Ref-Teacher to amplify this separability by aligning harmful prompts with the refusal feature and pushing harmless prompts away from it. This makes its **internal representations explicitly optimized for harmful-harmless discrimination**.
> Unlike standard safeguard models that rely on labeled harmful data, the Ref-Teacher leverages the model’s **intrinsic safety-related representation**, resulting in **stronger generalization** across diverse harmful datasets (Table 8).
> ## **Distributional Shifts (W3)**
> We would like to clarify that our data filtering **does not modify user samples, but only removes harmful samples.** As shown in Table 7, our filtering during finetuning is highly accurate, so any distributional shift from misclassification is **negligible in practice**. Even, when the user data contains harmful samples, removing them actually corrects the distribution toward the intended task, rather than distorting it, thereby improving both safety and utility.
> ## **Practicality of the Setting without a Pre-Aligned Model (Q4)**
> We **follow the same assumption used in prior works** such as Booster, Vaccine, and Lisa, which also start from a base LLM rather than assuming access to a safety-aligned model. This ensures a fair comparison with alignment-stage methods.
>
> This assumption is practical in scenarios where **new FaaS providers do not yet prepare a safety-aligned model**. Such organizations must decide how to construct one that is robust against harmful finetuning. They can choose (1) an alignment-stage solution, or (2) standard supervised safety-alignment with a finetuning-stage solution. In this setting, we propose a specialized safety-aligned model that accurately distinguishes harmful prompts.
>
> Importantly, **assuming that a safety-aligned model already exists is actually an easier setting.** If available, Ref-Teacher can be trained without updating the refusal feature or even replaced by the existing safety-aligned model (Table A8). Thus, while we target the more challenging scenario, **our framework naturally extends to cases where a safety-aligned model is already present.**
> ## **Impact of Filtered Data on Comparison (Q5)**
> We would like to clarify that **our method does not create a new safety-targeted dataset**; we **filter the user data** and finetune the base model jointly on the existing safety data and the filtered user data. While filtering improves safety, but **alignment distillation alone outperforms state-of-the-art methods** (Table 9).
>
> Nevertheless, to address your concern, **we will provide filtered-baseline results** in the camera-ready version.
> ## **Computational Cost (W1, Q6)**
> Although the reviewer requested FLOPs, they do not reflect actual training cost because they ignore backward-pass operations. Instead, we report GPUTime and GPUMemory for both the alignment and finetuning stages in Table R1, measured on four RTX 3090 GPUs.
>
> As shown in Table R1, our method incurs extra cost compared to SFT, but **the increase is not severe relative to other baselines**. Our method uses 44.0% more GPUTime and 35.9% more GPUMemory than SFT, yet it achieves a 93.8% reduction in harmful score and a 22.8% gain in finetuning accuracy. Thus, **the additional cost is justified by the substantial safety improvement.**

---

> > ### Comment · Reviewer_jpSC · 2025-11-26
> > **Response**
> >
> > Thank you for the clarifications.
> >
> > While they address some of my confusions about the proposed solution, the main concerns on the implementation complexity and expense overhead compared to state-of-the-art are not alleviated. While on paper it beats can beat others, practical adoption of this solution is questionable. I therefore keep my score.

---

> ### Author Response · Authors · 2025-11-27
>
> Dear Reviewer jpSC,
>
> Thank you for your follow-up comments. We are glad that some of your earlier concerns were clarified. However, we are concerned that the points about implementation complexity and computational overhead remain unresolved. This is unexpected, because our rebuttal already provided direct comparisons of computational costs against state-of-the-art baselines in **Table R1**.
>
> Table R1 shows that our method has **comparable cost** to existing approaches while achieving strong safety and utility. Moreover, prior methods incur substantially **larger overhead when they rely on external models such as LLaMAGuard for data filtering**, which is necessary for reliable Finetuning-as-a-Service. To make this point explicit, we additionally measure the cost of applying LLaMAGuard3-8B filtering to all baselines, reported in Table R10.
>
> **Table R10. Computational Cost including LLaMAGuard3-8B Data Filtering.**
> |Methods|Alignment GPUTime (s)|Alignment GPUMemory (GB)|Finetuning  GPUTime (s)|Finetuning  GPUMemory (GB)|Finetuning  GPUTime (s)|Finetuning  GPUMemory (GB)|
> |:-|:-:|:-:|:-:|:-:|:-:|:-:|
> |SFT|0.91|7.84|3.09|18.54|4.00|26.38|
> |Repnoise|4.27|15.27|3.09|18.54|7.36|33.81|
> |Vaccine|1.79|7.84|3.09|18.54|4.88|26.38|
> |Booster|3.92|9.39|3.09|18.54|7.01|27.93|
> |LDIFS|0.91|7.84|3.45|25.74|4.36|33.58|
> |Lisa|0.91|7.84|3.09|18.54|4.00|26.09|
> |Ref-Teacher (Ours)|1.76|11.29|1.84|12.01|3.60|23.30|
>
> As Table R10 shows, the cost of the finetuning stage increases considerably for baselines when external filtering is applied. In contrast, **our method maintains the same low cost because filtering is integrated into the Ref-Teacher** rather than implemented through external inference.
>
> We are willing to further discuss any remaining concerns during the discussion period. We will gladly clarify any part of the paper that seems unclear and will refine the explanation in the camera-ready version.
>
> Best regards,
> Authors of Paper 8215

---

### Author Response · Authors · 2025-11-21

We thank all reviewers for their thoughtful and constructive feedback. They highlighted the practical importance and relevance of the problem we study (Reviewer jpSC, Reviewer CsFX), the clarity and logical motivation of our formulation (Reviewer B6XU, Reviewer MKsJ), and the strong empirical performance showing clear improvements over state-of-the-art methods across diverse settings (Reviewer jpSC, Reviewer MKsJ). Reviewers also noted that the paper is well-written and easy to follow (Reviewer B6XU, Reviewer MKsJ), and that leveraging signals from a safety-aligned teacher is an interesting and meaningful direction (Reviewer CsFX). We sincerely appreciate these positive assessments. Responses to each reviewer’s concerns and questions have been added to their respective comment sections, and we will incorporate all clarifications, analyses, and suggested extensions into the revised manuscript and appendix. If anything remains unclear after reading our responses, please feel free to ask for further clarification at any time. Due to space limitations, all tables and references are placed here at the top of the reviews. All ambiguous points, additional analyses, and suggested extensions will be incorporated into the revised main manuscript and appendix.
## **Tables**
**Table R1. Computational Cost (average per-step, per-gpu).**
|Method|Alignment GPUTime (s)|Alignment GPUMemory (GB)|Finetuning GPUTime (s)|Finetuning GPUMemory (GB)|Sum GPUTime (s)|Sum GPUMemory (GB)|
|:-|:-:|:-:|:-:|:-:|:-:|:-:|
|SFT|0.91|7.84|1.59|9.31|2.50|17.15|
|RepNoise|4.27|15.27|1.59|9.31|5.86|24.58|
|Vaccine|1.79|7.84|1.59|9.31|3.38|17.15|
|Booster|3.92|9.39|1.59|9.31|5.51|18.70|
|LDIFS|0.91|7.84|1.95|16.51|2.86|24.35|
|Lisa|0.91|7.84|1.59|9.02|2.50|16.86|
|Ref-Teacher|1.76|11.29|1.84|12.01|3.60|23.30|

**Table R2. Comparison with SafeInstruct and Antidote on GSM8K.**
|Method|HS|FA|
|:-|:-:|:-:|
|SFT|16.7|40.6|
|RepNoise|30.4|38.4|
|Vaccine|4.8|24.4|
|Booster|5.9|43.4|
|LDIFS|4.0|17.0|
|SafeInstruct|7.0|41.5|
|Lisa|5.3|38.3|
|Antidote|21.6|45.8|
|Ref-Teacher|**0.5**|**49.0**|

**Table R3. Experiments on Llama3-8B-Instruct for GSM8K.** Ref-Teacher* denotes using the raw instruct model as the Ref-teacher without training, and Ref-Teacher denotes training instruct model under our method.
|Method|HS|FA|
|:-|:-:|:-:|
|SFT|64.0|66.0|
|LDIFS|15.9|66.8|
|SafeInstruct|26.9|66.4|
|Lisa|28.1|60.6|
|Antidote|17.4|59.3|
|Ref-Teacher*|13.9|65.8|
|Ref-Teacher|5.4|66.5|

**Table R4. Applying Ref-Teacher-guided Data Filtering to Baselines on GSM8K.**
|Method|HS|FA|
|:-|:-:|:-:|
|SFT|1.7|43.3|
|RepNoise|2.5|36.7|
|Vaccine|1.3|22.4|
|Booster|0.9|44.2|
|LDIFS|1.1|16.1|
|SafeInstruct|1.2|42.2|
|Lisa|1.3|38.5|
|Antidote|1.0|44.7|
|Ref-Teacher|**0.5**|**49.0**|

**Table R5. Results on Full Test Sets**
|Method|GSM8K HS|GSM8K FA|AGNEWS HS|AGNEWS FA|
|:-|:-:|:-:|:-:|:-:|
|SFT|16.7|39.42|28.2|82.45|
|RepNoise|30.4|33.06|58.6|85.86|
|Vaccine|4.8|21.76|29.5|85.07|
|Booster|5.9|43.44|5.3|84.88|
|LDIFS|4.0|15.92|12.5|73.59|
|SafeInstruct|7.0|40.11|7.4|86.72|
|Lisa|5.3|37.38|14.9|85.26|
|Antidote|2.2|46.02|77.5|35.34|
|Ref-Teacher|**0.5**|**47.92**|**1.2**|**86.83**|

**Table R6. Classification Accuracy (%) of Deep Ignorance during Finetuning Stage.**
|Datasets|Harmful|Harmless|Total|
|:-|:-:|:-:|:-:|
|GSM8K|68.68|95.67|89.29|
|SST2|31.31|100|93.19|
|AGNEWS|31.31|100|93.19|
|AlpacaEval|31.88|97.46|90.90|

**Table R7. Comparison with Previous Data-Filtering-based Methods on GSM8K.**
|Method|HS|FA|
|:-|:-:|:-:|
|SFT|16.7|40.6|
|RepNoise|30.4|38.4|
|Vaccine|4.8|24.4|
|Booster|5.9|43.4|
|LDIFS|4.0|17.0|
|Lisa|5.3|38.3|
|Deep Ignorance|13.7|42.2|
|Pharmacist|64.1|36.1|
|Ref-Teacher|**0.5**|**49.0**|

**Table R8. Classification Accuracy (%) of LlamaGuard-3-8B Finetuned on BeaverTails during Finetuning Stage**
|Datasets|Harmful|Harmless|Total|
|:-|:-:|:-:|:-:|
|GSM8K|100|97.89|98.10|
|SST2|100|100|100|
|AGNEWS|100|100|100|
|AlpacaEval|100|85.08|96.40|

**Table R9. Comparison of Filtered Baselines on GSM8K using LlamaGuard-3-8B Fine-Tuned on BeaverTails**
|Method|HS|FA|
|:-|:-:|:-:|
|SFT|2.2|42.6|
|Repnoise|2.7|38.5|
|Vaccine|1.3|21.9|
|Booster|1.3|43.6|
|LDIFS|1.3|16.6|
|Lisa|1.3|38.9|
|Ref-Teacher|**0.5**|**49.0**|

## **References**
[1] Arditi, Andy, et al. "Refusal in language models is mediated by a single direction." Advances in Neural Information Processing Systems 37 (2024): 136037-136083.

[2] Hsiung, Lei, et al. "Why LLM Safety Guardrails Collapse After Fine-tuning: A Similarity Analysis Between Alignment and Fine-tuning Datasets." arXiv preprint arXiv:2506.05346 (2025).

[3] Xiao, Yuxin, et al. "When Style Breaks Safety: Defending Language Models Against Superficial Style Alignment." arXiv preprint arXiv:2506.07452 (2025).

---

### Comment · Area_Chair_dn3V · 2025-11-26
**Please Review Author Response**

Dear Reviewers,

The authors have now responded to your comments. Could you review their response as soon as possible? If you have any further questions or concerns, please raise them as well.

Best,

Your AC

---

### Note · Authors · 2025-12-03

**Comment:**

The authors have decided to withdraw the manuscript from ICLR 2025. We sincerely thank the reviewers and the Area Chair for their valuable time and detailed comments. We will take the feedback into account to refine our work.

**Withdrawal Confirmation:**

I have read and agree with the venue's withdrawal policy on behalf of myself and my co-authors.